# Bright multicolor labeling of neuronal circuits with fluorescent proteins and chemical tags

Richi Sakaguchi[1,2,3], Marcus N Leiwe[1,3], Takeshi Imai[1,2,3]*

[1]Graduate School of Medical Sciences, Kyushu University, Fukuoka, Japan; [2]Graduate School of Biostudies, Kyoto University, Kyoto, Japan; [3]Laboratory for Sensory Circuit Formation, RIKEN Center for Developmental Biology, Kobe, Japan

**Abstract** The stochastic multicolor labeling method 'Brainbow' is a powerful strategy to label multiple neurons differentially with fluorescent proteins; however, the fluorescence levels provided by the original attempts to use this strategy were inadequate. In the present study, we developed a stochastic multicolor labeling method with enhanced expression levels that uses a tetracycline-operator system (Tetbow). We optimized Tetbow for either plasmid or virus vector-mediated multicolor labeling. When combined with tissue clearing, Tetbow was powerful enough to visualize the three-dimensional architecture of individual neurons. Using Tetbow, we were able to visualize the axonal projection patterns of individual mitral/tufted cells along several millimeters in the mouse olfactory system. We also developed a Tetbow system with chemical tags, in which genetically encoded chemical tags were labeled with synthetic fluorophores. This was useful in expanding the repertoire of the fluorescence labels and the applications of the Tetbow system. Together, these new tools facilitate light-microscopy-based neuronal tracing at both a large scale and a high resolution.

DOI: https://doi.org/10.7554/eLife.40350.001

*For correspondence:
t-imai@med.kyushu-u.ac.jp

**Competing interests:** The authors declare that no competing interests exist.

## Introduction

Neuronal circuits are the basis for brain function. Therefore, the reconstruction of neuronal wiring diagrams is key to understanding circuit function. Fluorescence imaging has been a powerful approach in visualizing the three-dimensional structure of neuronal morphology. In particular, fluorescent proteins are useful for labeling genetically-defined neuronal populations. In recent years, a number of tissue-clearing methods have been developed, and these have been optimized for use with fluorescent proteins and deep-tissue antibody staining (*Hama et al., 2011*; *Chung et al., 2013*; *Ke et al., 2013*; *Susaki et al., 2014*; *Richardson and Lichtman, 2015*). These new tools have expanded the scale of the available technologies for fluorescence imaging to whole-organ and whole-organism levels.

It is still difficult, however, to dissect and trace an individual neuron from a brain sample labeled with a single type of fluorescent protein. One way to overcome this problem is to improve the spatial resolution. Recently, we developed a tissue-clearing agent for high-resolution three-dimensional fluorescence imaging, named SeeDB2 (*Ke et al., 2013*). SeeDB2 was designed to minimize spherical aberrations, allowing for high-resolution imaging including super-resolution microscopy. In this approach, there was much improvement in the z-resolution, a critical factor for dissection of neuronal fibers crossing over along the z-axis. Similarly, expansion microscopy is also a promising new approach used to improve resolution in three-dimensional fluorescence imaging (*Chen et al., 2015*; *Ku et al., 2016*; *Tillberg et al., 2016*; *Chang et al., 2017*).

**eLife digest** The brain is made up of millions of cells called neurons, and it is important to learn how these neurons are wired together to better understand how the brain works. To make it easier to tell individual neurons apart in samples from brains, some scientists have developed a process called Brainbow that labels individual neurons with different fluorescent colors. Scientists have also created techniques called "tissue clearing" to make a brain transparent in the laboratory. These techniques make the brain see-through enough to allow scientists to study the wiring of the brain in three dimensions.

These multicolor labeling and tissue clearing techniques are very helpful for studying the brain. But they have an important limitation; the fluorescent colors are not bright enough to allow scientists to trace the long extensions called axons and dendrites that wire neurons together. As a result, tracing axons and dendrites was difficult and required cutting the brain into hundreds of thin slices. It could take several months for scientists to trace the path of a single neuron. Brighter fluorescent labeling colors would allow scientists to use high-powered microscopes to trace the entire length of a neuron in a whole brain much more quickly and easily.

Now, Sakaguchi et al. have developed a bright multicolor labeling method for neurons called Tetbow. Tetbow produces more vivid colors allowing scientists to trace the wiring of neurons over long distances in the mouse brain. Sakaguchi et al. combined Tetbow with tissue clearing techniques to dissect and trace many neurons in a whole mouse brain within a few days.

Neuroscientists can now use Tetbow to speed up the study of how neurons are wired in the brain. Researchers working in other fields could also use Tetbow to help track the behavior of different cells. Tetbow allows everyone to see the beautiful wiring of the brain in three dimensions.
DOI: https://doi.org/10.7554/eLife.40350.002

Another approach to the dissection of neuronal circuits is multicolor labeling. To facilitate the dissection of individual neurons, a transgenic multicolor labeling method, Brainbow, has been developed, in which three different fluorescent proteins were expressed in a stochastic manner (*Livet et al., 2007*; *Cai et al., 2013*; *Loulier et al., 2014*). Brainbow used the Cre-loxP system to express one of the three fluorescent protein genes stochastically in a transgene. When multiple copies of the transgene cassette are introduced, stochastic choices will result in a combinatorial expression of these three genes with different copy numbers, producing dozens of color hues.

Although the Brainbow concept is powerful for discriminating between numerous neurons using light microscopy, the existing Brainbow methods are of limited use for neuronal tracing. This is because the stochastic and combinatorial expression of fluorescent proteins is possible only at low copy number ranges for the transgenes, so that the expression levels of the fluorescent proteins were not sufficiently high for bright and high-resolution imaging of axons and dendrites. Therefore, many of the previous studies were forced to use subsequent antibody staining to produce reliable neuronal tracing. In the present study, we utilized the Tet-Off system (Tetbow) to develop a multicolor labeling method with enhanced expression. As vector (plasmid and virus)-mediated gene transfer has become a versatile tool in modern neuroscience, we aimed to perform multicolor labeling using these tools. As a proof-of-concept experiment, we demonstrated the ability to trace axons of individual neurons on the scale of several millimeters in the mouse olfactory system. To improve the stability of the fluorescence labels after harsh tissue-clearing treatment, we also developed a Tetbow system with chemical tags. When combined with the advances in the growing field of tissue-clearing techniques, these new multicolor labeling strategies should facilitate neuronal tracing at higher densities and resolutions.

## Results

### A trade-off between expression levels and color variation

Earlier Brainbow methods utilized transgenic animals for stochastic multicolor labeling. They utilized the Cre-loxP system, in which DNA recombination resulted in a stochastic selection of one fluorescent protein gene out of three (or more) choices. When multiple copies of the Brainbow transgene

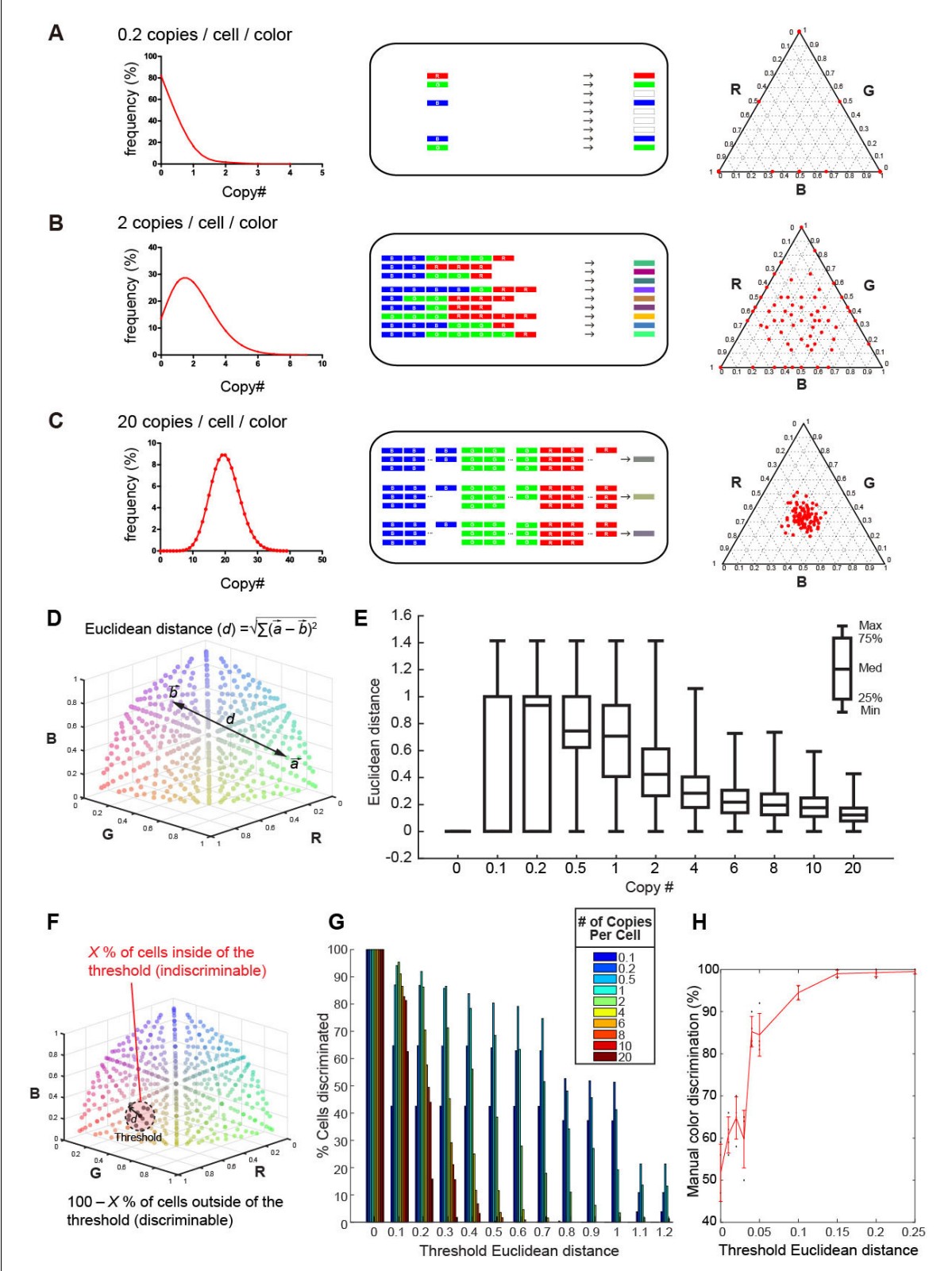

**Figure 1.** Tetbow strategy. Stochastic representation of the vector-mediated multicolor labeling strategy. Here we assume that the copy numbers of introduced vectors follow a Poisson distribution. In a plasmid or virus-mediated gene transfer method, we do not need to use the Cre-loxP system. Rather, it is important to limit the number of genes introduced into each cell. For example, in theory, an average of 2 copies/cell/color can result in the stochastic expression of three different genes (**B**). If we introduce a copy number that is too small (e.g. an average of 0.2 copies/cell; **A**), only one of the

*Figure 1 continued on next page*

*Figure 1 continued*

three genes will be expressed in most cells. If too many genes are introduced in each cell (e.g. an average of 20 copies/cell; C), color variations will be reduced because many of the neurons will express a similar number of XFP genes. The use of the Cre-loxP system alone cannot solve this problem. Simulations of the expected color variations that are based on the various Poisson distributions (middle) are shown as ternary plots (100 plots/condition, right panel). The numerical data are presented in *Figure 1—source data 1*. Note that many of the plots are overlapping at the edge of the triangle in (A). Three different XFPs are shown in red (R), green (G), and blue (B). (D) The difference between two colors represented as Euclidean distance in 3D color space. For example, a difference between colors *a* and *b* is represented as the distance, *d*. (E) Box plots of Euclidean distances in all pairs of plots in 3D space in relation to copy number. The horizontal lines within each box represents the median, the box represents the interquartile range, and the whiskers show the minimum and maximum values. (F) A cartoon illustrating the cells inside (*X*%) and outside the threshold Euclidean distance. The cells outside (100 − *X* %) are considered to be discernable. (G) The percentage of discernable cells for each threshold *d* and each copy number. Simulation data used for (E) and (G) are provided in *Figure 1—source data 1*. (H) The color discrimination abilities of experienced researchers. Tests are explained in *Figure 1—figure supplement 1*. Mean ± SD are indicated in red. When threshold *d* is 0.1, the score was 94.5 ± 1.73% (mean ± SD). Score data are in *Figure 1—source data 2*.

DOI: https://doi.org/10.7554/eLife.40350.003

The following source data and figure supplement are available for figure 1:

**Source data 1.** Simulation data used for *Figure 1E, G*.

DOI: https://doi.org/10.7554/eLife.40350.005

**Source data 2.** Color discrimination scores in *Figure 1H*.

DOI: https://doi.org/10.7554/eLife.40350.006

**Figure supplement 1.** Explanation of human color discrimination tests.

DOI: https://doi.org/10.7554/eLife.40350.004

were introduced into the genome, stochastic recombination produced a variety of color hues based on different copy numbers of expressed fluorescent protein genes (collectively called XFPs) (*Livet et al., 2007*). In recent years, however, plasmid or virus vector-mediated gene transfer has become a more versatile strategy in neuroscience. We therefore tried to optimize a multicolor labeling method for vector-mediated gene transfer.

Previously, an adeno-associated virus (AAV)-mediated Brainbow method (AAV-Brainbow) has been reported (*Cai et al., 2013*). However, in the vector-mediated gene transfer, the Cre-loxP system is not essential for the stochastic and combinatorial expression of XFP genes (*Kobiler et al., 2010*; *Weber et al., 2011*; *Siddiqi et al., 2014*). We can easily make color variations by introducing a mixture of three different XFP constructs: as long as the copy number of the introduced genes is small, labeled neurons will still produce a variety of colors irrespective of whether the Cre-loxP system is used or not (*Figure 1A–C*). It should be noted, however, that color variation reduces as the copy number of the introduced genes increases. We can estimate the optimum copy number of the introduced XFP genes as follows. We considered that the number of introduced genes will follow a Poisson distribution (*Kobiler et al., 2010*). When three different XFP genes are introduced at 20 copies/cell/color on average, a similar number of copies will be introduced into each neuron, and only a small degree of color variation will be generated (*Figure 1C*). By contrast, if the copy number is too small, many of the neurons will express just one XFP gene (*Figure 1A*). When these three genes are introduced at an average 2 copies/cell/color, much larger color variations will be produced (*Figure 1B*).

We wanted to determine the optimum copy number of the expressed XFP genes on the basis of this simulation. We plotted the color values for cells in the color-coding space after intensity normalization (total intensity = 1). In this coding space, each dimension represents the intensity of one of the three colors (Red, Green, and Blue in pseudocolor representation). The mean Euclidean distance (*d*) for two randomly chosen cells was greater when the copy number was lower (*Figure 1D, E*), but this does not necessarily mean that we can discriminate between many cells, as the two cells are more likely to become the same color when copy number is too low (*Figure 1A*). We therefore calculated the probability that two randomly chosen cells are discriminated on the basis of a given threshold distance in the color-coding space. Here we considered that cells within the threshold distance (*d*) from a reference are indiscriminable in the color-coding space; cells outside of the threshold distance were considered discriminable from the reference (*Figure 1F*). In our simulation, when we assumed a threshold distance of 0.1, 95.3% of cells could be discriminated from a given cell when XFP genes were expressed at 2 copies/color/cell (*Figure 1G*). We also found that experienced

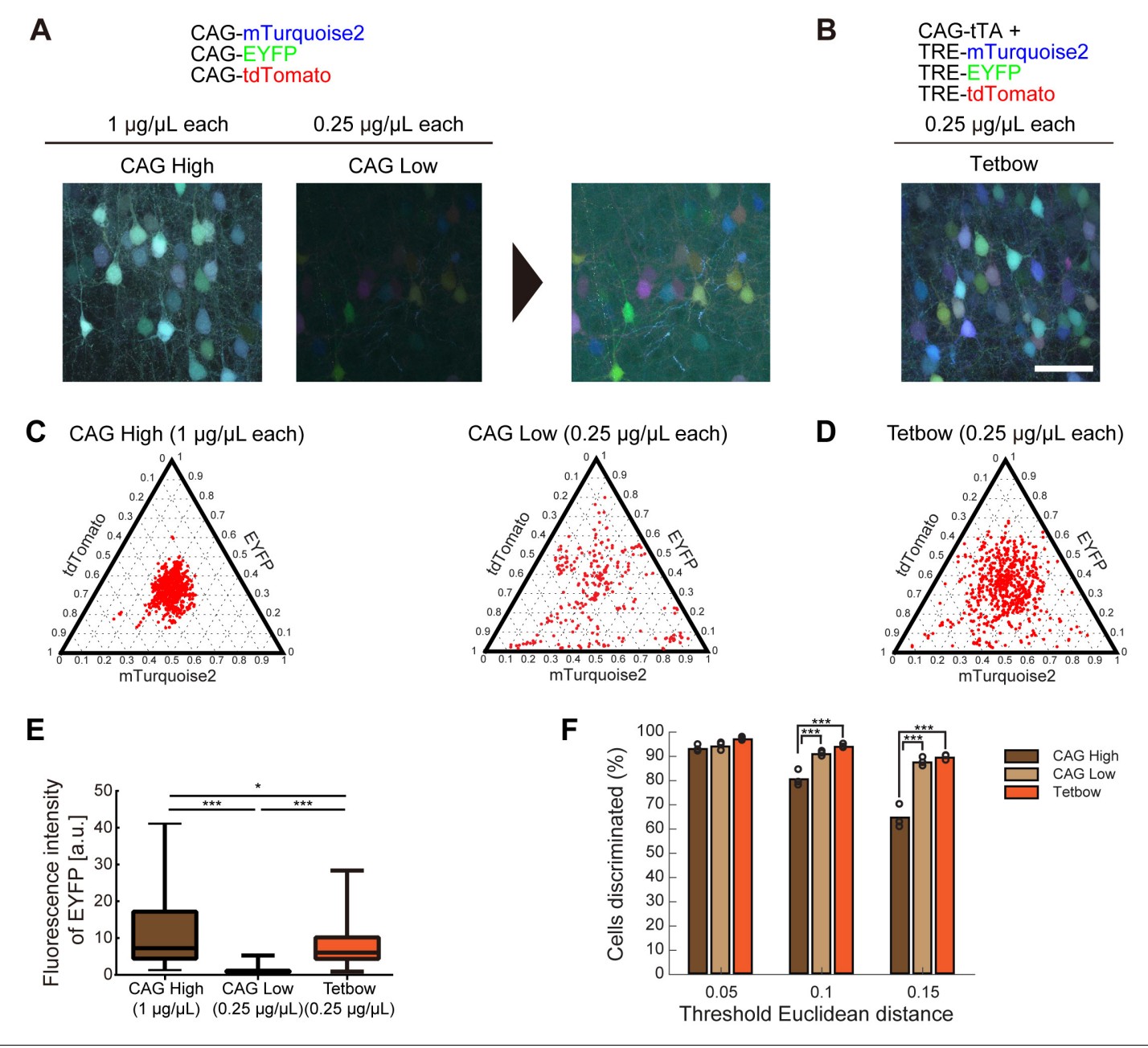

**Figure 2.** Bright multicolor labeling can be produced by using fewer copy numbers of plasmids with a stronger promoter. (**A**) L2/3 neurons labeled with CAG-Turquoise2, EYFP, and tdTomato (1 or 0.25 µg/µL each). When each plasmid was introduced at 0.25 µg/µL, the color variation was increased, but the fluorescence levels were reduced. Enhanced images are shown on the right. (**B**) CAG-tTA, TRE- Turquoise2, EYFP, and tdTomato were introduced at 0.25 µg/µL each (Tetbow construct; see *Figure 2—figure supplement 1*). *In utero* electroporation was performed at E15 and the mice were analyzed at P21. Experiments were performed in parallel, and image acquisition conditions were the same for (**A**) and (**B**). Scale bars represent 50 µm. (**C, D**) Color variations made by three CAG vectors (**C**) vs. Tetbow vectors (**D**) are shown in ternary plots. Fluorescence intensities were vector normalized and the fluorescence intensity ratios were compared. Fluorescence intensities at neuronal somata were used for the quantitative comparison. Note that the color variations are small with high-copy CAG vectors, whereas color variations are high with the Tetbow vectors (n = 249–728 cells per group from three sets of experiments for each of the samples). Fluorescence intensity data used for (**C**), (**D**), and (**F**) are provided in *Figure 2—source data 1*. (**E**) Boxplot of median EYFP fluorescence intensities (normalized to the median of CAG (0.25 µg/µl). A D'Agostino and Pearson Normality Test showed that the data were not normally distributed (p<0.0001). EYFP signals in (**A**) and (**B**) were compared. The horizontal line in each box represents the median location, the box represents the interquartile range, and the whiskers show the minimum and maximum values. p*<0.05, p***<0.001 (Kruskal-Wallis with Dunn's post hoc test, n = 104–276 per sample). The data are from one set of experiments performed in parallel. We analyzed three independent sets of experiments and obtained similar results. The fluorescence intensity data used for (**E**) are in *Figure 2—source data 2*. (**F**) Percentage of discernable

*Figure 2 continued on next page*

*Figure 2 continued*

cells in the condition of each threshold *d* and each copy number. p***<0.001 (two-way ANOVA with Tukey-Kramer post hoc test, n = 249–728 per sample).

DOI: https://doi.org/10.7554/eLife.40350.007

The following source data and figure supplements are available for figure 2:

**Source data 1.** Fluorescence intensity data used for C,*Figure 2C*, *D* and *F*.

DOI: https://doi.org/10.7554/eLife.40350.010

**Source data 2.** Fluorescence intensity data used for *Figure 2E*.

DOI: https://doi.org/10.7554/eLife.40350.011

**Figure supplement 1.** Tetbow constructs.

DOI: https://doi.org/10.7554/eLife.40350.008

**Figure supplement 2.** Comparison with Brainbow.

DOI: https://doi.org/10.7554/eLife.40350.009

researchers can discriminate two colors separated by 0.1 Euclidean distances in the color-coding space at 94.5 ± 1.73% accuracy (mean ± S.D.; *Figure 1H* and *Figure 1— figure supplement 1*). Thus, ~2 copies/color/cell is the optimum when the color hues are judged visually by human experimenters.

We evaluated this prediction using the *in utero* electroporation of plasmid vectors into layer 2/3 cortical pyramidal neurons (electroporated at embryonic day 15). We introduced a mixture of three separate plasmid vectors encoding mTurquoise2 (blue), EYFP (green), and tdTomato (red) genes under a CAG promoter. When these plasmids were expressed at high copy numbers (CAG High; DNA solution introduced was 1 µg/µL/plasmid; 3 µg/µL in total; see 'Materials and methods' section for details), only small color variations were produced (*Figure 2A,C*). When the DNA concentrations were reduced (CAG Low; 0.25 µg/µL/plasmid), the color variation increased (*Figure 2B*), but the overall fluorescence levels were much reduced (*Figure 2E*). Thus, there is a trade-off between the expression levels of fluorescent proteins and color variations, and this is the reason why the previous Brainbow methods were unable to produce color variations that were bright enough for labeling.

## Tetbow: multicolor labeling with enhanced expression levels

We therefore wanted to enhance the expression levels of fluorescent proteins while maintaining the required low copy numbers of XFP genes. Recent studies indicated that the tetracycline response element (TRE) promoter ensures much higher expression levels than the CAG promoter when expressed with a tetracycline trans-activator (tTA) (*Madisen et al., 2015*; *Sadakane et al., 2015*). We therefore used the tTA-TRE (Tet-Off) system instead of a common CAG promoter. We also introduced a Woodchuck hepatitis virus posttranscriptional regulatory element (WPRE) sequence in the 3′′ UTR of XFP genes to improve their expression levels. Earlier Brainbow techniques used membrane-bound XFPs because unmodified XFPs labeled the somata too brightly and affected the tracing of nearby neurites (*Cai et al., 2013*). However, this problem can be easily solved by minimizing spherical aberration in microscopy by using an index-matched clearing agent, SeeDB2 (*Ke et al., 2016*). Furthermore, unmodified XFPs label axons and dendrites much more brightly than the membrane-bound ones (data not shown). We therefore used unmodified XFPs instead of membrane-bound XFPs. These new set of constructs (named Tetbow; *Figure 2—figure supplement 1*) achieved much higher XFP expression levels when compared with the CAG-promoter plasmids (*Figure 2B*). When we quantified the expression levels of EYFP, we observed a six-fold increase in fluorescence levels (*Figure 2E*; CAG 0.25 µg/µL vs. Tetbow 0.25 µg/µL; n = 104 and 260, respectively, p<0.0001, Kruskal Wallis with Dunn's post hoc test). Tetbow allowed for much more robust multicolor labeling than the Brainbow constructs when introduced by *in utero* electroporation, allowing more reliable axon tracing (*Figure 2—figure supplement 2*).

Owing to the enhanced expression, the Tetbow system achieved bright labeling (*Figure 2E*) while maintaining the color variations produced by low copy numbers of XFP genes. When CAG-XFP genes were introduced at high concentrations (1 µg/µL each), the color variations were small; by contrast, the Tetbow constructs (0.25 µg/µL) produced much larger variations (*Figure 2D*). In the color discrimination analysis, we confirmed that different cells are better discriminated by Tetbow than in the CAG High condition (*Figure 2F*).

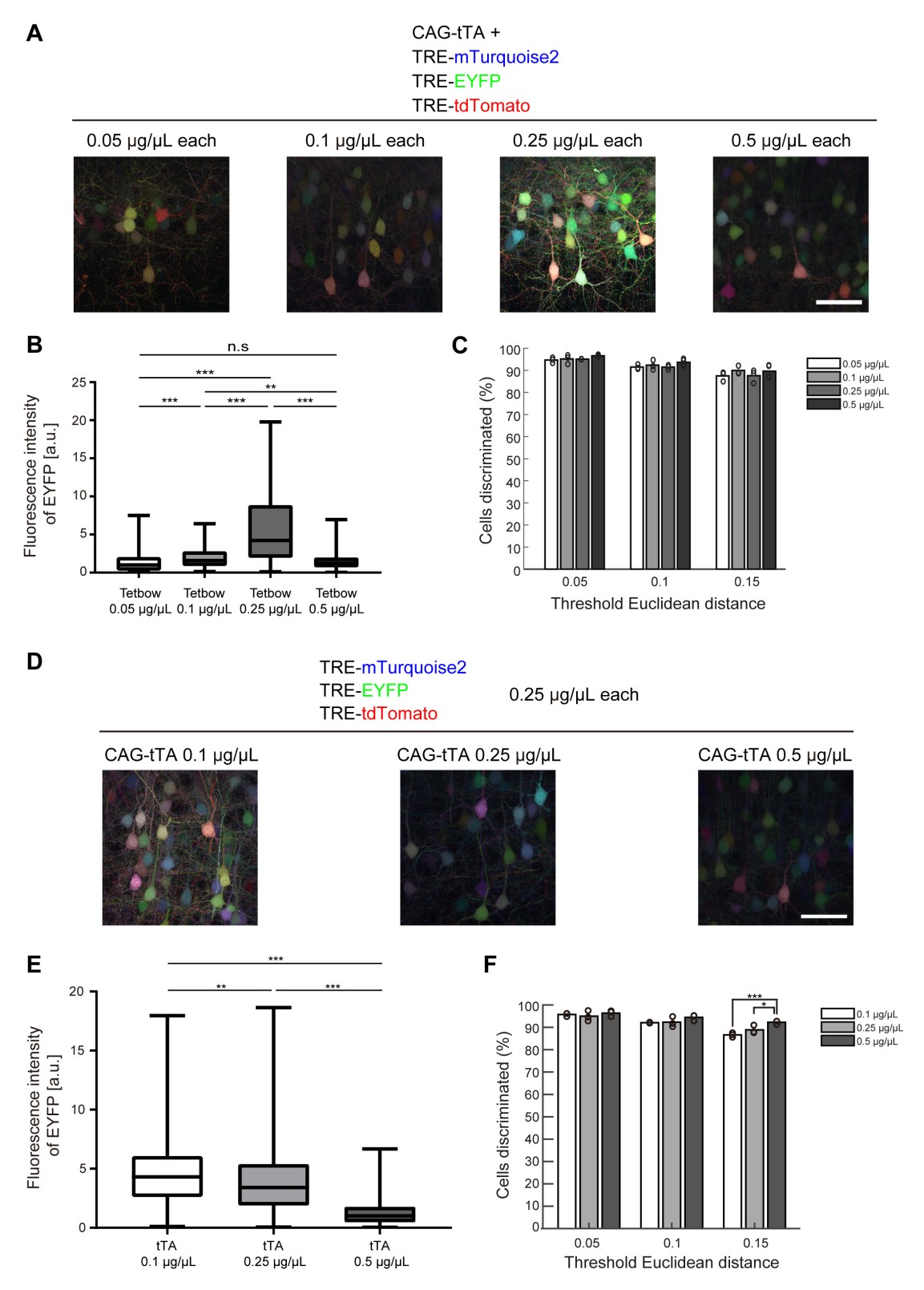

**Figure 3.** Optimization of Tetbow plasmid concentrations for *in utero* electroporation. (**A–C**) Plasmid concentrations for the four Tetbow vectors (CAG-tTA and three TRE-XFP vectors) were tested at 0.05, 0.1, 0.25, and 0.5 μg/μL. (**B**) Boxplot of median EYFP fluorescence intensities (normalized to the median of Tetbow (0.05 μg/μl)). A D'Agostino and Pearson normality test showed that the data were not normally distributed (p<0.0001). The horizontal bar within each box represents the median fluorescence intensity, the box represents the interquartile range, and the whiskers show the minimum and
*Figure 3 continued on next page*

*Figure 3 continued*

maximum values. p**<0.01, p***<0.001 (Kruskal Wallis with Dunn's post hoc test, n = 273–524 per sample). (C) Percentage of discernable cells for each of the threshold *d* and dilution conditions. There was no significant difference between each *d* condition (two-way ANOVA with Tukey-Kramer post hoc test) (n = 273–524 per sample). The fluorescence intensity data used for (B) and (C) are available in *Figure 3—source data 1*. (D–F) L2/3 neurons labeled with Tetbow. Only tTA concentrations were changed (0.1, 0.25, or 0.5 µg/µL). Decrease in tTA concentrations enhanced signal intensities. (E) Boxplot of median EYFP fluorescence intensities (normalized to the median of Tetbow (tTA: 0.5 µg/µl)). A D'Agostino and Pearson normality test showed that the data were not normally distributed (p<0.0001). The horizontal bar within each box represents the median location, the box represents the interquartile range, and the whiskers show the minimum and maximum values. p**<0.01, p***<0.001 (Kruskal Wallis with Dunn's post hoc test, n = 356–414 per sample). (F) Percentage of discernable cells for each threshold *d* and each dilution condition. p*<0.05, p***<0.001 (two-way ANOVA with Tukey-Kramer post hoc test, n = 356–414 per sample). The fluorescence intensity data used for (E) and (F) are available in *Figure 3—source data 2*.

DOI: https://doi.org/10.7554/eLife.40350.012

The following source data and figure supplement are available for figure 3:

**Source data 1.** Source data for *Figure 3B,C*.
DOI: https://doi.org/10.7554/eLife.40350.014
**Source data 2.** Source data for *Figure 3E,F*.
DOI: https://doi.org/10.7554/eLife.40350.015
**Figure supplement 1.** Tetbow optimization.
DOI: https://doi.org/10.7554/eLife.40350.013

We further tried to determine the optimum concentration of plasmids for Tetbow when labeled with *in utero* electroporation. Among the four plasmid concentrations we tested (0.05, 0.1, 0.25, and 0.5 µg/µL each), color discrimination performance was comparable. Paradoxically, however, the expression levels of XFP were the highest at 0.25 µg/µL each, not at 0.5. We therefore considered the possibility that a moderate expression level of tTA is critical for the optimum expression. When 0.1, 0.25, and 0.5 µg/µL of CAG-tTA plasmid was co-introduced with 0.25 µg/µL each of TRE-XFP plasmids, the highest expression level was found with 0.1 µg/µL CAG-tTA (*Figure 3D–F*). Thus, too high a concentration of tTA leads to the suppression of TRE-XFP genes. Consistent with this assumption, when the WPRE sequence was added to the CAG-tTA plasmid, the expression level of TRE-XFP genes was reduced (*Figure 3—figure supplement 1*). Thus, the expression level of tTA needs to be moderate to achieve the highest expression of TRE-XFP genes.

## High-resolution 3D imaging with tissue clearing

We described the use of SeeDB2 as a tissue-clearing agent (*Ke et al., 2016*). SeeDB2 is designed to minimize spherical aberrations for glycerol (SeeDB2G) or oil (SeeDB2S) immersion objective lenses, making high-resolution 3D imaging possible. Furthermore, various fluorescent proteins were best preserved in SeeDB2, much more so than in commercialized mounting media or other tissue-clearing agents (*Ke et al., 2016*). In high-resolution imaging, we must obtain photons from a limited volume, and thus the fluorescence intensity must be sufficiently high. Therefore, the combination of Tetbow and SeeDB2 is ideal for high-resolution volumetric multicolor imaging.

We introduced Tetbow constructs into Layer 2/3 neurons in the cerebral cortex using *in utero* electroporation. After clearing with SeeDB2G, the fluorescence levels of XFPs with Tetbow were strong enough to allow visualization of the fine detail of neuronal morphology in 3D (*Figure 4A* and *Video 1*). When we analyzed adult brain slices (P70), detailed structures of dendritic spines (*Figure 4B*, *Figure 4—figure supplement 1*, and *Video 2*) and axonal boutons (*Figure 4C*) were clearly visualized with Tetbow. It should be noted that we obtained these high-resolution images of synaptic structures using solely native fluorescence of XFPs, with no antibody staining.

To evaluate the versatility of Tetbow, we also tested other types of neurons using *in utero* electroporation. For example, we were able to label brightly mitral and tufted (M/T) cells in the olfactory bulb with Tetbow. It is known that each glomerulus in the olfactory bulb is innervated by 20–50 M/T cells (*Ke et al., 2013*; *Imai, 2014*). When we looked at each glomerulus, dendrites from different mitral cells were clearly visualized and were distinguishable by their different colors (*Figure 5* and *Video 3*). Thus, Tetbow can be used to analyze the detailed dendrite wiring diagram of individual M/T cells, including 'sister' M/T cells, which are connected to the same glomerulus.

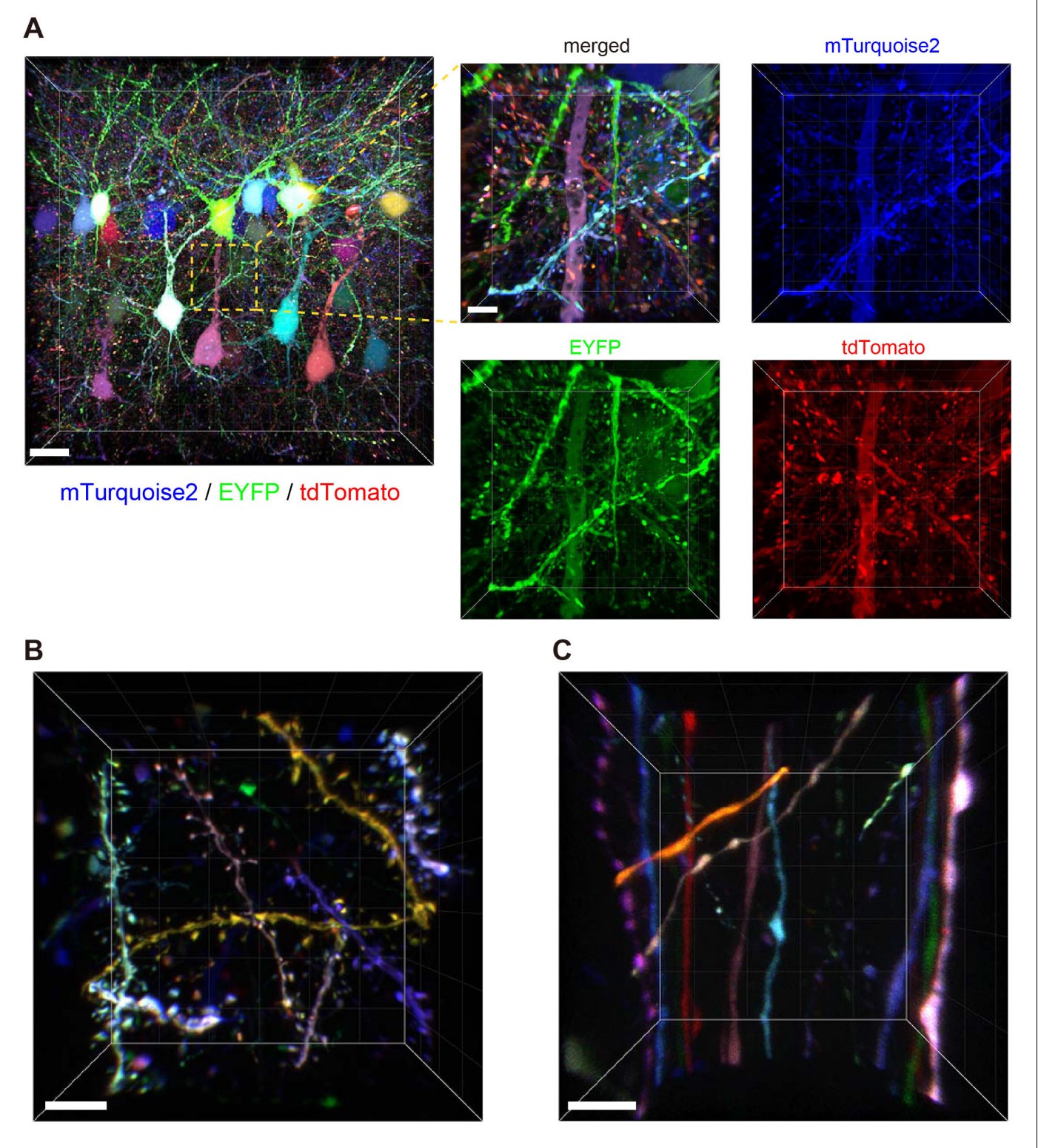

**Figure 4.** Tetbow labeling is bright enough for high-resolution imaging of synaptic structures. Volume rendering of Layer 2/3 cortical pyramidal neurons labeled with Tetbow (P70). *In utero* electroporation was used to label L2/3 neurons at E15. Brain slices were cleared with SeeDB2G and imaged with confocal microscopy. Different neurons were brightly labeled with different colors. Representative images are shown from four independent experiments. (A) Low- and high-magnification images in Layer 2/3 (45.16 μm thick). The four panels on the right indicate each of the three single channel fluorescence images. See also *Video 1*. (B) Dendrites and dendritic spines were brightly labeled with various colors with Tetbow. A volume-
*Figure 4 continued on next page*

*Figure 4 continued*

rendered image (18.65 µm thick) is shown. See also *Video 2*. (**C**) Axons and axonal boutons were clearly visualized with Tetbow. A volume rendered image (23.01 µm thick) is shown. Note that the synaptic-scale structures were clearly visualized with the native fluorescence of XFPs, without antibody staining. Scale bars are 20 µm (A, left) and 5 µm (A, right and B, **C**).

DOI: https://doi.org/10.7554/eLife.40350.016
The following figure supplement is available for figure 4:

**Figure supplement 1.** Tetbow labeling is bright enough in thick brain slices.
DOI: https://doi.org/10.7554/eLife.40350.017

## Tetbow with chemical tags

Aqueous tissue-clearing agents are useful for large-scale three-dimensional imaging with fluorescent proteins. However, to clear lipid-rich myelinated axons completely, harsh clearing treatments, such as the use of detergents, solvents, and heating, are inevitable (*Dodt et al., 2007*; *Chung et al., 2013*; *Renier et al., 2014*; *Susaki et al., 2014*; *Murray et al., 2015*). Indeed, there is a trade-off between the transparency of the tissues and damage to the tissues and fluorescent proteins (*Ke et al., 2013*; *Ke et al., 2016*). For example, most of the solvent-based clearing methods, such as BABB, quench fluorescent proteins. To overcome this problem, chemical tags could be a promising alternative to fluorescent proteins (*Kohl et al., 2014*; *Sutcliffe et al., 2017*). Genetically encoded chemical tags, such as SNAP, Halo, CLIP, and TMP form covalent bonds with their cognate substrate, and fluoresce with synthetic labels. The molecular weights of these ligands are relatively small, allowing easy penetration into thick tissues. Once they form covalent bonds with their substrates, the fluorescence remains stable even under harsh clearing conditions.

We tested Tetbow multicolor labeling with three different chemical tags, SNAP, Halo, and CLIP (*Figure 6—figure supplement 1*). We introduced these three chemical tag genes into L2/3 cortical pyramidal neurons using *in utero* electroporation. Brains were fixed, and the three chemical tags were visualized with synthetic fluorescence labels: SNAP-Surface 488, HaloTag TMR Ligand, and CLIP-Surface 647, respectively (*Figure 6A*). Like fluorescent proteins, chemical tags allowed for the robust multicolor labeling of neurons using the Tetbow system (*Figure 6B*). Owing to the low molecular weight of the substrates, 1 mm-thick mouse brain samples were efficiently labeled (*Figure 6—figure supplement 2*). After tissue clearing with solvent-based tissue clearing agents, 3DISCO or BABB, fluorescent proteins were largely quenched (data not shown) (*Ke et al., 2013*); synthetic fluorophores bound to chemical tags, however, were bright and stable under these clearing conditions (*Figure 6B*).

## Tetbow AAVs

AAVs are also becoming a versatile gene delivery tool in neuroscience. However, the size of the conventional Brainbow gene cassettes was too large to allow them to be packaged into an AAV vector (<5 kb). For this reason, a previous study divided the Brainbow cassette into two separate AAVs with two XFP genes each, and employed Cre-loxP recombination (*Cai et al., 2013*). As described above, however, the Cre-loxP system is not needed to achieve multicolor labeling using AAV-mediated gene expression. Using the Tetbow strategy (i.e. multiple AAV vectors with different XFP genes), we can solve the size problem, and achieve improved multicolor labeling using simplified DNA constructs.

We generated four separate AAV vectors carrying *SYN1*-tTA, TRE-mTurquoise2, TRE-EYFP, and TRE-tdTomato genes (*Figure 7—figure supplement 1*). The human *SYN1* promoter was used to express tTA specifically in neurons. These four

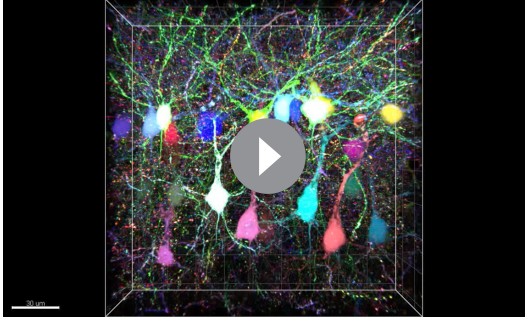

**Video 1.** Layer 2/3 cortical pyramidal neurons labeled with Tetbow. Tetbow plasmids were introduced at E15 and the cerebral cortex was analyzed at P70. See legends to *Figure 4A*.
DOI: https://doi.org/10.7554/eLife.40350.018

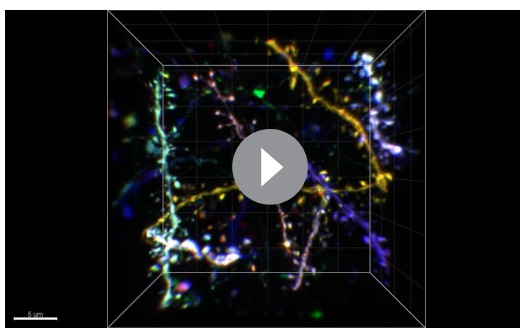

**Video 2.** High-magnification images of dendrites in Layer 2/3 cortical pyramidal neurons labeled with Tetbow. Tetbow plasmids were introduced at E15 and the cerebral cortex was analyzed at P70. See legends to *Figure 4B*.
DOI: https://doi.org/10.7554/eLife.40350.019

AAV vectors (serotype AAV2/1) were injected into the cerebral cortex of adult mice (P56) and analyzed two weeks later. We found a stochastic and combinatorial expression of the encoded fluorescent proteins in Layer five neurons when injected at $4 \times 10^8$ gc/mL of AAV-*SYN1*-tTA and $3 \times 10^{10}$ gc/mL of AAV-TRE-XFPs (*Figure 7A*). As expected from our simulation (*Figure 1*), the virus titer was critical for producing color variations. Higher virus titers led to reduced color variations (*Figure 7B*, right). In the cerebral cortex, 1–$3 \times 10^{10}$ gc/mL of AAV-TRE-XFPs was found to be optimum on the basis of the resultant expression levels and color variations (*Figure 7C and D*), but the optimum range may be different for different cell types (*Figure 7—figure supplement 2*), injection volumes, or virus serotypes. Expression levels were sufficiently high for high-resolution imaging of neuronal morphology, including the visualization of dendritic spines (*Video 4*). It should be noted, however, that the expression of Tetbow AAVs can lead to toxic effects on cellular functions after a prolonged incubation period, as a result of the extremely high levels of expression. For example, cortical neurons started to show an aggregation of XFPs and displayed morphological abnormalities 4 weeks after virus injection, whereas olfactory bulb neurons were best visualized at 4 weeks. Thus, the optimum incubation time may be different for different cell types.

## Long-range axon tracing with Tetbow

The bright multicolor labeling method, Tetbow, is particularly useful for the analysis of long-range axonal projection of a population of neurons. To test the utility of Tetbow, we focused on mitral and tufted (M/T) cells in the olfactory bulb, which project axons (up to several millimeters) to the olfactory cortex, including the anterior olfactory nucleus, olfactory tubercle, piriform cortex, cortical amygdala, and lateral entorhinal cortex. Previous studies have performed axon tracing on populations of M/T cells from glomeruli in the olfactory bulb using dye injections (*Nagayama, 2010*; *Sosulski et al., 2011*), but these studies could not fully dissect individual axons. Other studies performed single-cell axon tracing using hundreds of serial brain sections, but these analyses were highly laborious and time-consuming (*Ghosh et al., 2011*; *Igarashi et al., 2012*). Owing to these limitations, we do not yet understand fully how the odor inputs into individual M/T cells are conveyed to the olfactory cortex.

To examine the axonal projections of M/T cells from the olfactory bulb, we injected Tetbow AAVs into the mouse olfactory bulb. First, Tetbow AAVs were expressed in most of olfactory bulb neurons. When compared to single-color labeling, the stochastic and combinatorial expression of XFPs was helpful in improving tracing performance (*Figure 8—figure supplement 1*). Efficient axon tracing was further facilitated by the local injection of Tetbow AAVs (*Figure 8A,B*). In the olfactory bulb, many types of neurons were labeled, but only M/T cells projected their axons to the olfactory cortex. To facilitate high-resolution fluorescence imaging with a limited working distance of the objective lens (20x, NA = 0.75, WD = 0.66 mm), we flattened the olfactory cortical area onto a glass slide (*Sosulski et al., 2011*). After the fixation and tissue clearing, axons of individual M/T cells were clearly visualized with XFPs in the entire area of the olfactory cortex (*Figure 8D* and *Video 5*). The unique color hue was largely preserved at the average level, if not at a single-pixel resolution, from proximal to distal part in each axon (*Figure 8E*). Across the three image areas, the median Euclidean distance for the same neuron was 0.049 (interquartile range = 0.063, n = 34 pairs). The unique color hues and their consistency facilitated manual reconstruction of individual M/T cell axons in the olfactory cortex (*Figure 8F*, *Figure 8—figure supplement 2*, and *Video 6*). Highly divergent patterns of axonal collaterals were observed among labeled M/T cell axons, complementing earlier findings (*Nagayama, 2010*; *Ghosh et al., 2011*; *Sosulski et al., 2011*).

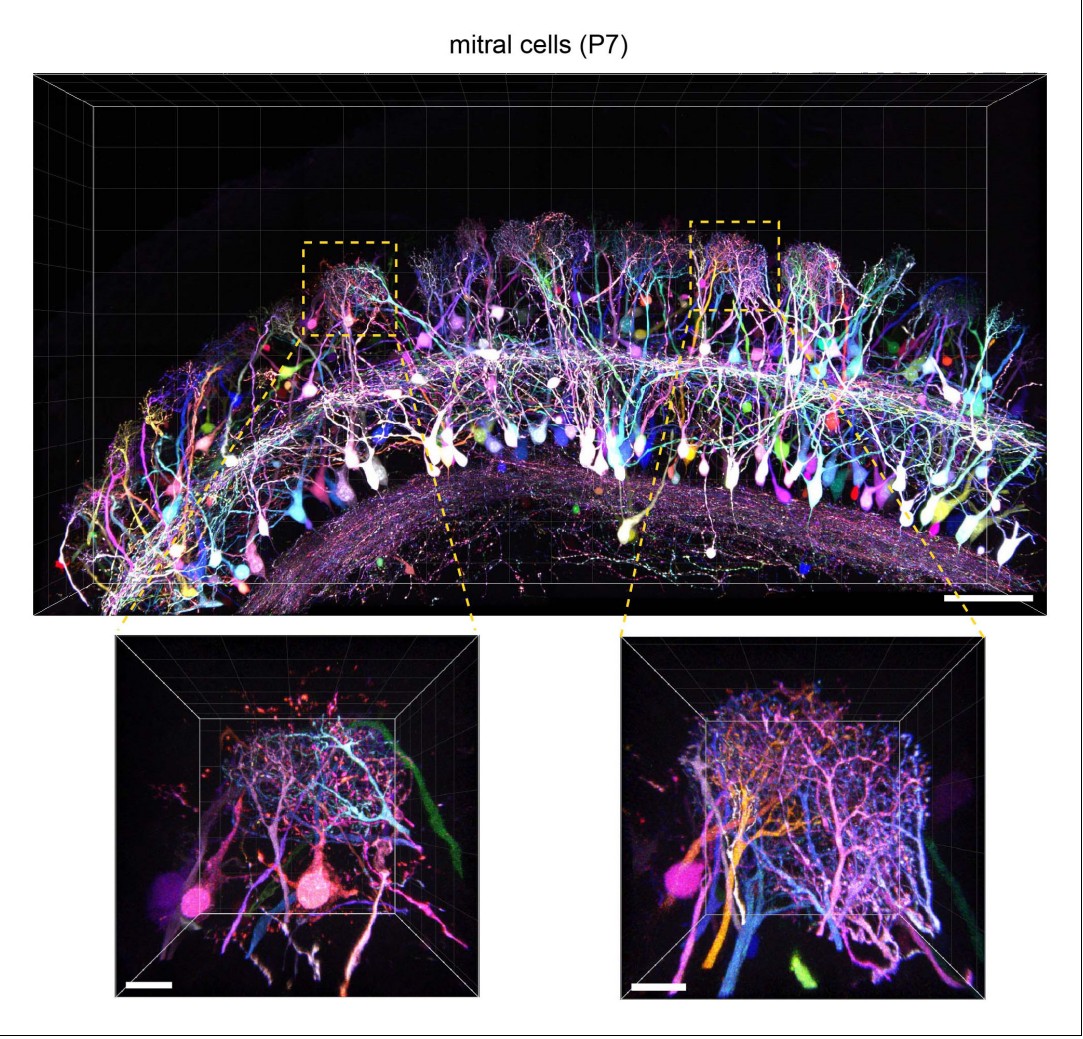

mitral cells (P7)

**Figure 5.** Tetbow labeling in the olfactory bulb. M/T cells in the olfactory bulb (P7) were labeled with Tetbow. A volume-rendered image is shown (84.07 μm thick). Each mitral cell was labeled with unique colors facilitating the identification of individual neurons at high resolution. Note that dendritic tufts from different neurons are clearly distinguishable with different colors in the high-magnification image (bottom). Representative data from one out of three independent experiments are shown here. See also *Video 3*. Scale bars are 100 μm (top) and 20 μm (bottom). Fluorescence signals at M/T cell somata are intentionally saturated in this image so that the fine details of dendrites are better visualized.
DOI: https://doi.org/10.7554/eLife.40350.020

## Discussion

### Bright multicolor labeling of neurons using the Tet-Off system

To achieve the stochastic expression of multiple fluorescent proteins, it is important to optimize the number of genes that are introduced per cell. We introduced a limited number of copies of plasmids or virus vectors into neurons to enable the stochastic expression of XFPs based on a Poisson distribution. To enhance the expression levels of XFPs, we utilized a Tet-Off system, in which the tetracycline trans-activator (tTA) binds to the TRE promoter to drive the expression of

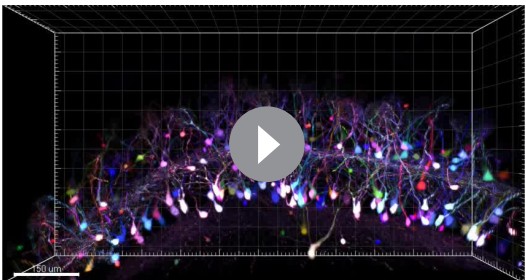

**Video 3.** Mitral cells of the olfactory bulb labeled with Tetbow. Tetbow plasmids were introduced at E12 and the olfactory bulb samples were analyzed at P7. See legends to *Figure 5*.
DOI: https://doi.org/10.7554/eLife.40350.021

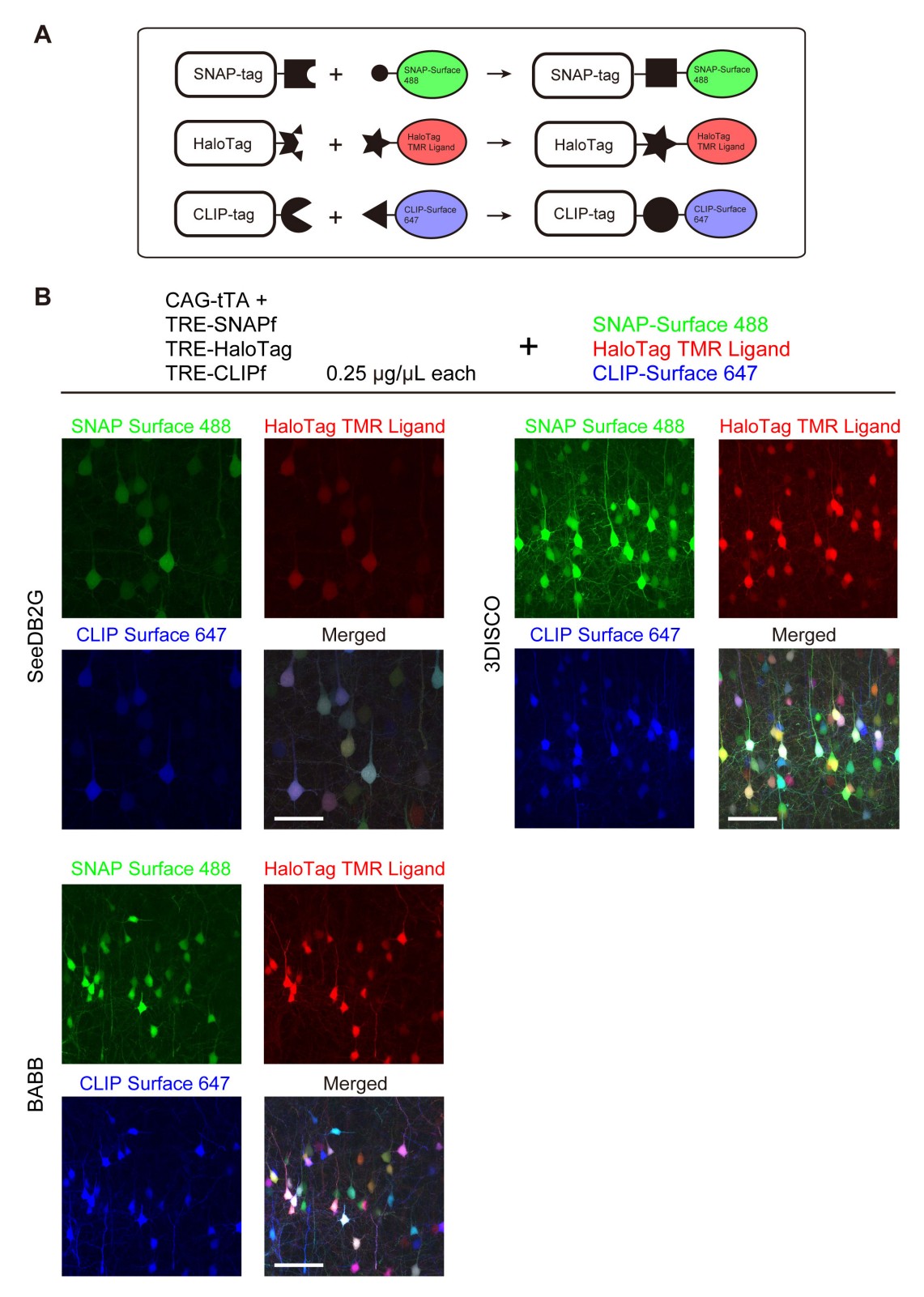

**Figure 6.** Tetbow labeling with chemical tags. (**A**) Tetbow labeling with chemical tags. We used SNAP, CLIP, and Halo tags for Tetbow labeling. These chemical tags form covalent bonds with their substrates, which contain chemical fluorophores. These chemical labels are stable even under harsh clearing conditions. See *Figure 6—figure supplement 1* for plasmid construction. (**B**) L2/3 neurons labeled with SNAP, Halo, and CLIP tags were visualized with SNAP-Surface 488, HaloTag TMR Ligand, and CLIP-Surface 647 (P21), respectively. Brain tissue was cleared with SeeDB2G, BABB, or

*Figure 6 continued*

3DISCO. Note that a fluorescent protein, mTurquoise2, was largely quenched by BABB and 3DISCO whereas chemical tags with synthetic dyes remained stable. Scale bars are 50 µm. Data for each condition were obtained from sections from the same animal. This process was replicated for three animals.

DOI: https://doi.org/10.7554/eLife.40350.022

The following figure supplements are available for figure 6:

**Figure supplement 1.** Plasmid maps for chemical-tag Tetbow.

DOI: https://doi.org/10.7554/eLife.40350.023

**Figure supplement 2.** Efficient labeling of thick brain slices with chemical tags.

DOI: https://doi.org/10.7554/eLife.40350.024

genes of interest. In this way, we achieved multicolor labeling with much-improved brightness. Previously, the electroporation and virus infection of the Brainbow construct introduced just a small number of genes per cell, and as a result, the fluorescence levels were not sufficiently high for reliable neuronal tracing (*Kobiler et al., 2010*; *Egawa et al., 2013*) (see also *Figure 2—figure supplement 2*). In fact, Brainbow methods often had to employ antibody staining to enhance the fluorescence signals (*Cai et al., 2013*). However, our Tetbow strategy provides enhanced fluorescence, allowing for high-resolution imaging without the need for antibody staining. As a result, our Tetbow strategy has opened up a new opportunity for large-scale high-resolution neuronal tracing using tissue clearing.

Our Tetbow strategy also overcomes a size-limit problem associated with AAV vectors. As we introduced tTA and TRE-XFP gene cassettes with separate AAV vectors, DNA construction was also simplified. Using our Tetbow AAVs, we could clearly visualize different neurons with different color hues at synaptic resolution. It should be noted, however, that using the correct virus titer is critical for the generation of color variations. If the virus titer is too low, many of the labeled neurons will express just one XFP gene. By contrast, if the virus titer is too high, many of the labeled neurons will express all three genes at similar levels. Recently, another group employed a similar strategy using a Tet-Off system and a newly engineered AAV for intravenous transduction; but the brightness levels that they achieved were not as good as those provided by CAG vectors (*Chan et al., 2017*). To achieve the highest expression levels of XFPs, the expression level of tTA needs to be optimized (*Figure 3*). The Tet-Off system is also useful for controling the sparseness of XFP expression; by simply diluting the tTA vector, the expression of XFP can be sparsened without affecting color variations (*Chan et al., 2017*).

AAV- based Tetbow is useful for a relatively broad set of applications, including use in less genetically tractable model animals. For example, an AAV-based Tet-Off system has already been tested in the marmoset brain (*Sadakane et al., 2015*). Like mice, these animals demonstrated high expression levels of fluorescent proteins when the Tet-Off system was used. Thus, our Tetbow strategy can be useful for neuronal tracing studies in various species including primates.

## Multicolor labeling with chemical tags

In the present study, we also extended the Tetbow method to include chemical tags. Current tissue clearing methods are intended for the use of fluorescent proteins, but chemical tags can become a good alternative to the fluorescent proteins. First, unlike fluorescent proteins, once labeled with synthetic fluorophores, chemical tags are stable under harsh clearing conditions. This means that we can have a broader choice of clearing methods, particularly for lipid-rich and thick tissues. Chemical tags may also be useful for expansion microscopy, where the stability of fluorescent proteins has been a challenging aspect (*Chen et al., 2015*; *Ku et al., 2016*; *Tillberg et al., 2016*; *Chang et al., 2017*). The fluorescence labeling of chemical tags is much easier than the antibody staining of XFPs (*Kohl et al., 2014*). Furthermore, synthetic fluorophores penetrate much deeper than antibodies because of their smaller size. Second, more choices of fluorescence spectrum and photochemical properties are available with chemical tags than with fluorescent proteins. For example, synthetic fluorophores cover the spectrum from UV to near-IR range, expanding the possible spectrum range when imaging. In addition, autonomously blinking fluorophores are very useful for localization-based super-resolution microscopy (*Uno et al., 2014*). Multicolor 3D PALM/STORM imaging of cleared tissues would be an interesting possibility in the future.

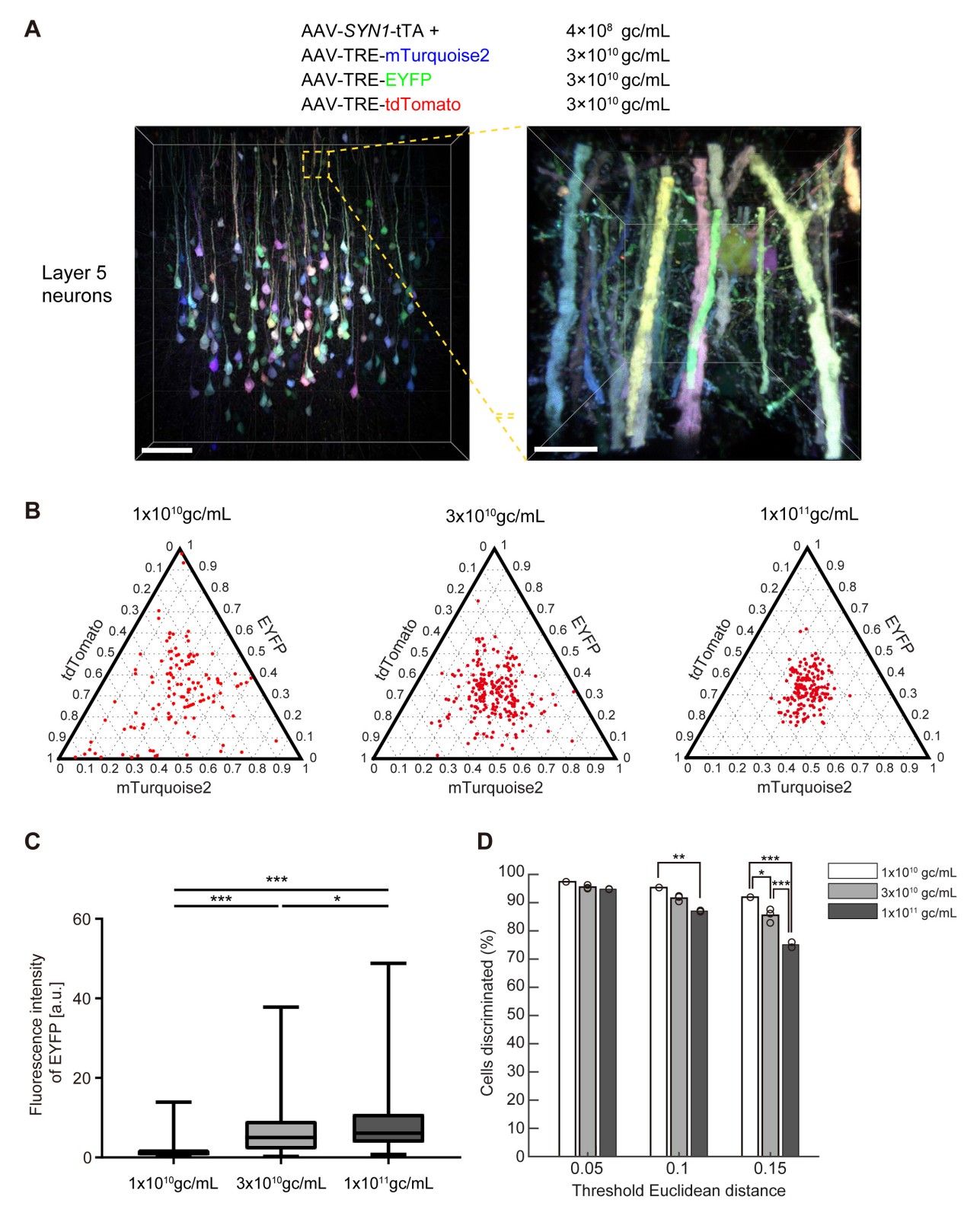

**Figure 7.** Tetbow labeling with AAVs. (**A**) Layer five pyramidal neurons in the cerebral cortex labeled with Tetbow AAVs. Tetbow AAVs (serotype AAV2/1) were injected at P56 and analyzed two weeks later. Brain slices were cleared with SeeDB2G. The images display the native fluorescence of XFPs under SeeDB2G clearing. Low- and high-magnification images (89 μm thick, volume rendering) are shown. Scale bars are 100 μm on the left, and 10 μm on the right. (**B**) The optimization of virus titer conditions. Various titers of TRE-mTurquoise2, TRE-EYFP and TRE-tdTomato ($1 \times 10^{10}$, $3 \times 10^{10}$, $1 \times 10^{11}$

*Figure 7 continued*

gc/mL) were injected at P56 and analyzed two weeks later. The tTA titer was $4 \times 10^8$ gc/mL throughout. (C) Boxplots of EYFP fluorescence intensities at the cell bodies (normalized to the median of $1 \times 10^{10}$ gc/mL). A D'Agostino and Pearson normality test showed that the data were not normally distributed (p<0.0001). The horizontal line within each box represents the median location, the box represents the interquartile range, and the whiskers represent the minimum and maximum values. p*<0.05, p***<0.001 (Kruskal-Wallis with Dunn's post hoc test, n = 163–220 per sample). (D) The percentage of discernable cells for each threshold $d$ and each titer condition. p*<0.05, p**<0.01, p***<0.001 (two-way ANOVA with Tukey-Kramer post hoc test, n = 163–220 per sample). Fluorescence intensity data used for (B) to (D) are available in *Figure 7—source data 1*.

DOI: https://doi.org/10.7554/eLife.40350.025

The following source data and figure supplements are available for figure 7:

**Source data 1.** Fluorescence intensity data used for *Figure 7*B-D.

DOI: https://doi.org/10.7554/eLife.40350.028

**Figure supplement 1.** AAV vector maps.

DOI: https://doi.org/10.7554/eLife.40350.026

**Figure supplement 2.** Various brain areas labeled with Tetbow AAVs.

DOI: https://doi.org/10.7554/eLife.40350.027

## Single-axon tracing for long-range axonal projections

In recent years, several tissue-clearing methods that are optimized for fluorescence imaging have been developed, expanding the imaging scale in light-microscopy-based neuronal tracing. However, the imaging resolution was not sufficiently high for densely labeled neuronal circuits. Therefore, we could only look at population-level connectivity (*Oh et al., 2014*) or at a very small subset of neurons (*Economo et al., 2016*) with light microscopy. To examine the projection diagram of individual neurons, a single-cell barcoding and RNA sequencing approach, MAPseq, has been proposed (*Kebschull et al., 2016*). Although MAPseq is a promising new tool in the study of area-to-area connectivity for hundreds of neurons (*Han et al., 2018*), we cannot know the finer details of their neuronal morphology. Currently, saturated connectomics is only possible with electron microscopy (*Kasthuri et al., 2015*), but the analytical throughput is currently not sufficient for the study of long-range projections. The combination of Tetbow and tissue clearing can become a useful tool to fill the gap, allowing the dissection of densely labeled neurons at a large scale.

## Limitations and future challenges

In the present study, we employed our Tetbow method to analyze axonal projection profiles for M/T cells in the mouse olfactory bulb (*Figure 8*). We were able to find labeled axons in all the areas of the olfactory cortices that we examined (piriform cortex, cortical amygdala, and lateral entorhinal cortex), but it remains unclear whether we could completely label all the fibers to their termini. In fact, it is extremely difficult to know whether the labeled termini are really termini or are interrupted by small unlabeled gaps. Even if the second scenario is very likely, it is difficult to know which segments are in fact connected to each other, but the variable color hues can help. Single-cell labeling is advantageous to avoid such ambiguity, although the throughput is limited for such analyses (*Igarashi et al., 2012*). By contrast, multicolor labeling may be useful to compare the projection patterns in the same animals. In our analysis shown in *Figure 8F*, we traced axons using conservative criteria: we terminated tracing when labeled axons were interrupted by small gaps. Nevertheless, the quality of axonal tracing was comparable to that in earlier studies (*Nagayama, 2010*; *Ghosh et al., 2011*). To further improve the tracing performance (including up to the axon termini), it will be important to improve labeling methods in order to fill the thin neuronal fibers completely and evenly.

In this study, we employed manual neuronal tracing to Tetbow data, and were able to trace individual M/T cell axons successfully at the millimeter (>6 mm) scale. Color hues were consistent at a global scale (*Figure 8E*), but not perfect in high-magnification images (*Figure 8D*). Chromatic aberration can also be a problem in the high-resolution imaging of thick tissues. Another important challenge will be to improve the color hue consistency throughout a neuron and to develop auto-tracing software that is optimized for the multicolor-labeled neurons.

# Materials and methods

**Key resources table**

| Reagent type (species) or resource | Designation | Source or reference | Identifiers | Additional information |
|---|---|---|---|---|
| Gene (*Mus musculus*) | C57BL/6N | Japan SLC | RRID: MGI:5658686 | |
| Gene (*Mus musculus*) | ICR | Japan SLC | RRID: MGI:5652524 | |
| Cell line (*Homo sapiens*) | AAVpro 293 T Cell Line | Clontech | cat# 632273 | |
| Recombinant DNA reagent | pTRE-Tight | Clontech | cat# 631059 | |
| Recombinant DNA reagent | pCAG-CreERT2 | PMID: 17209010 (*Matsuda and Cepko, 2007*) | Addgene# 14797 | |
| Recombinant DNA reagent | pmTurquoise2-N1 | PMID: 22434194 (*Goedhart et al., 2012*; *Matsuda and Cepko, 2007*) | Addgene# 60561 | |
| Recombinant DNA reagent | pBluescript II SK(+) phagemid | Agilent Technologies | cat# 212205 | |
| Recombinant DNA reagent | paavCAG-pre-mGRASP-mCerulean | PMID: 22138823 (*Kim et al., 2011*) | Addgene# 34910 | |
| Recombinant DNA reagent | pSNAPf | New England Biolabs | cat# N9183S | |
| Recombinant DNA reagent | pCLIPf | New England Biolabs | cat# N9215S | |
| Recombinant DNA reagent | pFC14K HaloTag CMV Flexi Vector | Promega | cat# G3780 | |
| Recombinant DNA reagent | pCAG-mTurquoise2 | This paper | N/A | |
| Recombinant DNA reagent | pCAG-EYFP | This paper | N/A | |
| Recombinant DNA reagent | pCAG-tdTomato | This paper | N/A | |
| Recombinant DNA reagent | pCAG-tTA | This paper | Addgene# 104102 | |
| Recombinant DNA reagent | pBS-TRE-mTurquoise2-WPRE | This paper | Addgene# 104103 | |
| Recombinant DNA reagent | pBS-TRE-EYFP-WPRE | This paper | Addgene# 104104 | |
| Recombinant DNA reagent | pBS-TRE-tdTomato-WPRE | This paper | Addgene# 104105 | |
| Recombinant DNA reagent | pBS-TRE-SNAPf-WPRE | This paper | Addgene# 104106 | |
| Recombinant DNA reagent | pBS-TRE-CLIPf-WPRE | This paper | Addgene# 104107 | |
| Recombinant DNA reagent | pBS-TRE-HaloTag-WPRE | This paper | Addgene# 104108 | |
| Recombinant DNA reagent | AAV2-miniSOG-VAMP2-tTA-mCherry | PMID: 23889931 (*Lin et al., 2013*) | Addgene# 50970 | |
| Recombinant DNA reagent | pAAV-*SYN1*-tTA | This paper | Addgene# 104109 | |
| Recombinant DNA reagent | pAAV-TRE-mTurquoise2-WPRE | This paper | Addgene# 104110 | |
| Recombinant DNA reagent | pAAV-TRE-EYFP-WPRE | This paper | Addgene# 104111 | |

*Continued on next page*

*Continued*

| Reagent type (species) or resource | Designation | Source or reference | Identifiers | Additional information |
|---|---|---|---|---|
| Recombinant DNA reagent | pAAV-TRE -tdTomato-WPRE | This paper | Addgene# 104112 | |
| Recombinant DNA reagent | pCAG-iCre | PMID: 26972009 (*Ke et al., 2016*) | N/A | |
| Recombinant DNA reagent | AAV-EF1a-BbTagBY | PMID: 23817127 (*Cai et al., 2013*) | Addgene# 45185 | |
| Recombinant DNA reagent | AAV-EF1a-BbChT | PMID: 23817127 (*Cai et al., 2013*) | Addgene# 45186 | |
| Commercial assay or kit | AAVpro Helper Free System | Takara | cat# 6673 | |
| Commercial assay or kit | AAVpro Purification Kit (All Serotypes) | Takara | cat# 6666 | |
| Commercial assay or kit | AAVpro Titration Kit (for real-time PCR) | Takara | cat# 6233 | |
| Chemical compound, drug | SNAP-Surface 488 | New England Biolabs | cat# S9124S | |
| Chemical compound, drug | CLIP-Surface 647 | New England Biolabs | cat# S9234S | |
| Chemical compound, drug | HaloTag TMR Ligand | Promega | cat# G8252 | |
| Chemical compound, drug | Saponin | Nakalai-tesque | cat# 30502–42 | |
| Chemical compound, drug | Omnipaque 350 | Daiichi-Sankyo | cat# 081–106974 | |
| Chemical compound, drug | Urea | Wako | cat# 219–00175 | |
| Chemical compound, drug | N,N,N',N'-Tetrakis (2-hydroxypropyl) ethylenediamine | TCI | cat# T0781 | |
| Chemical compound, drug | Triton X-100 | Nakalai-tesque | cat# 12967–45 | |
| Chemical compound, drug | Hexane | Nakalai-tesque | cat# 17922–65 | |
| Chemical compound, drug | Benzyl alcohol | SIGMA | cat# 402834–100 ML | |
| Chemical compound, drug | Benzyl benzoate | Wako | cat# 025–01336 | |
| Chemical compound, drug | Tetrahydrofuran, super dehydrated, with stabilizer | Wako | cat# 207–17905 | |
| Chemical compound, drug | Dibenzyl Ether | Wako | cat# 022–01466 | |
| Software, algorithm | ImageJ | NIH | RRID: SCR_003070 | |
| Software, algorithm | GraphPad Prism 7 | GraphPad Software | RRID: SCR_002798 | |
| Software, algorithm | MATLAB | MathWorks | RRID: SCR_001622 | |
| Software, algorithm | Leica Application Suite X | Leica Microsystems | RRID: SCR_013673 | |
| Software, algorithm | Imaris | Bitplane AG | RRID: SCR_007370 | |

*Continued on next page*

*Continued*

| Reagent type (species) or resource | Designation | Source or reference | Identifiers | Additional information |
|---|---|---|---|---|
| Software, algorithm | Neurolucida | MBF Bioscience | RRID: SCR_001775 | |
| Software, algorithm | Adobe Photoshop | Adobe | RRID: SCR_014199 | |
| Software, algorithm | Adobe Illustrator | Adobe | RRID: SCR_010279 | |
| Other | Microslicer | Dosaka EM | PRO7N | |
| Other | Electroporator | BEX | CUY21EX | |
| Other | Forceps-type electrodes (5 mm diameter) | BEX | LF650P5 | |
| Other | Forceps-type electrodes (3 mm diameter) | BEX | LF650P3 | |
| Other | TCS SP8X | Leica Microsystems | TCS SP8X | |
| Other | 10x dry lens | Leica Microsystems | HC PL APO 10x/0.40 CS | |
| Other | 20x multi-immersion objective lens | Leica Microsystems | HC PL APO 20x /0.75 IMM CORR CS2 | |
| Other | 40x oil-immersion objective lens | Leica Microsystems | HC PL APO 40x OIL CS2, | |
| Other | 63x glycerol-immersion objective lens | Leica Microsystems | HC PL APO 63x GLYC CORR CS2 | |
| Other | Raw image data | This paper | http://ssbd.qbic.riken .jp/set/20180901/ | |
| Other | Neurolucida reconstruction data | This paper | http://ssbd.qbic.riken. jp/set/20180901/ | |
| Other | MATLAB codes | This paper | https://github.com/ mleiwe/TetbowCodes | |

## Plasmids

pCAG-mTurquoise2/EYFP/tdTomato were assembled by using pCAG-CreERT2 (Addgene #14797, RRID: Addgene_14797, from Dr. Cepko), pmTurquoise2-N1 (Addgene #60561, RRID: Addgene_60561, from Dr. D. Gadella), EYFP (Clontech), and tdTomato (a gift from Dr. R. Tsien). To generate a backbone vector for pTRE-XFP, an *Xho-Xho* fragment containing TRE-SV40 poly A was transferred from pTRE-Tight (#631059, Clontech) to pBluescript II SK(+) (# 212205, Agilent) to ensure high-copy expression in *Escherichia coli*. The WPRE sequence was PCR amplified from an aavCAG-pre-mGRASP-mCerulean vector (Addgene #34910, RRID: Addgene_34910, a gift from Dr. J. Kim). The tTA sequence in pCAG-tTA and pAAV2-*SYN1*-tTA was derived from the tTA2 section of the pTet-Off Advanced vector (Clontech). SNAP$_f$ and CLIP$_f$ tag genes were obtained from New England Biolabs (#N9183S, #N9215S), and the HaloTag gene was obtained from Promega (#G3780). AAV2-Tetbow vectors were generated by modifying AAV2-miniSOG-VAMP2-tTA-mCherry (Addgene #50970, RRID: Addgene_50970, from Dr. R. Tsien). Maps for the Tetbow

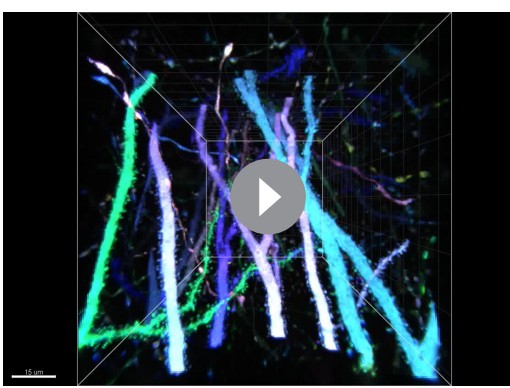

**Video 4.** CA1 pyramidal neurons in the hippocampus labeled with Tetbow AAVs. Tetbow AAVs were injected to the hippocampus at P56 and analyzed after two weeks. See legends to *Figure 7—figure supplement 2*.
DOI: https://doi.org/10.7554/eLife.40350.029

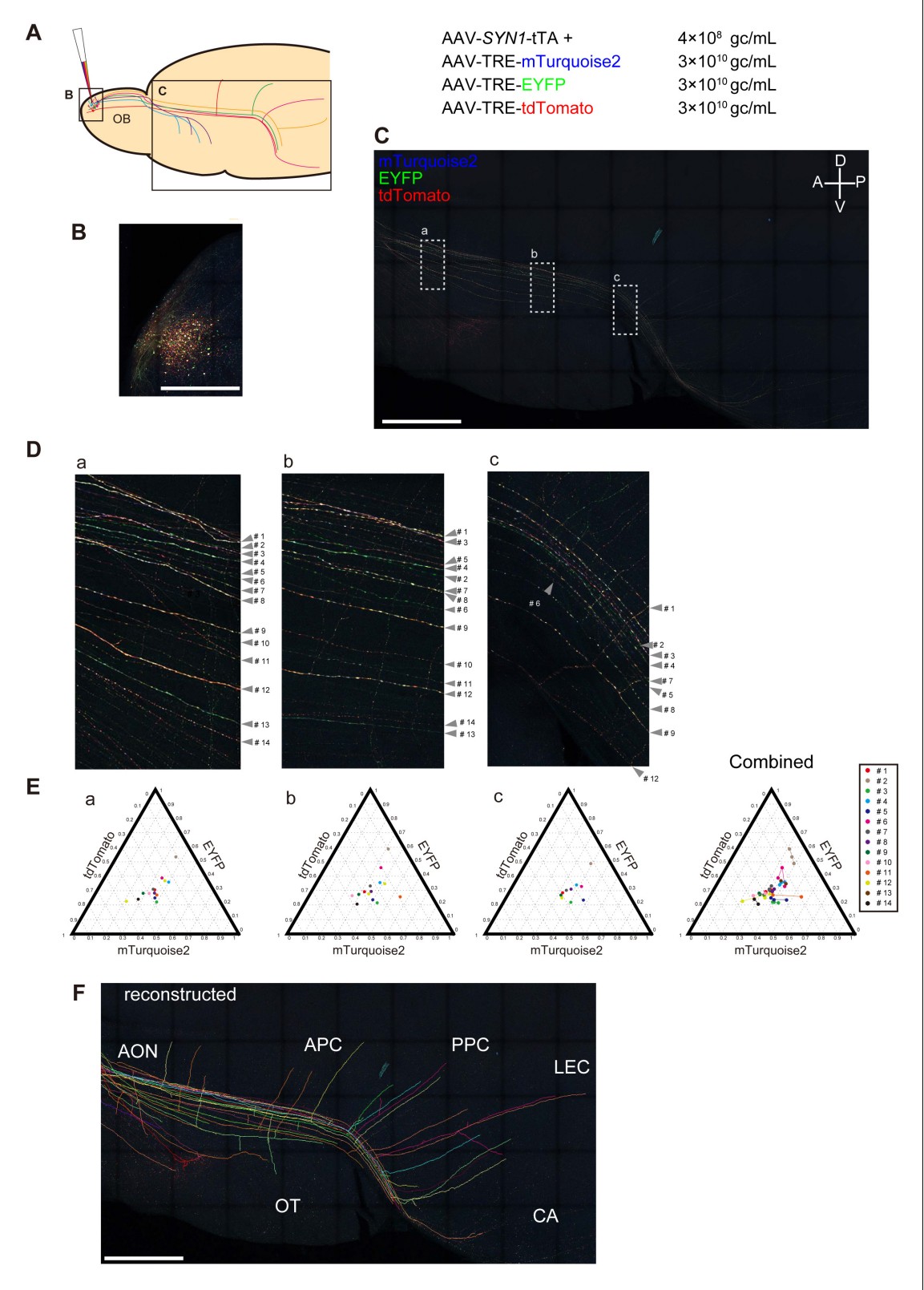

**Figure 8.** Long-range tracing of M/T cell axons with Tetbow AAVs. (**A**) M/T cells in the olfactory bulb were labeled with Tetbow AAVs. The Tetbow AAV cocktail (138 nL) was injected into a single location on the dorsal surface of the right olfactory bulb using a glass capillary. The virus was injected at P60 and mice were analyzed four weeks later. AAV titers were $4 \times 10^8$ gc/mL for AAV2/1-*SYN1*-tTA2 and $3 \times 10^{10}$ gc/mL each for AAV2/1-TRE-mTurquoise2-WPRE, AAV2/1-TRE-EYFP-WPRE, and AAV2/1-TRE-tdTomato-WPRE. (**B**) In the olfactory bulb, M/T cells and various types of interneurons

*Figure 8 continued on next page*

Figure 8 continued

were densely labeled with Tetbow. (**C**) In the olfactory cortex, only M/T cell axons were labeled with Tetbow AAVs. The ventral-lateral part of the brain was flattened to 700 µm thickness. After fixation and clearing, confocal images were taken with 20x objective lenses. The labeling density was lower in the olfactory cortex, so we could easily trace individual M/T cell axons. M/T cell axons were found in the all areas of the olfactory cortex, including anterior olfactory nucleus (AON), olfactory tubercle (OT), piriform cortex (anterior (APC) and posterior (PPC)), cortical amygdala (CA), and lateral entorhinal cortex (LEC). Maximal intensity projection images of confocal images are shown (379.89 µm). See also *Video 5*. (**D**) High-magnification versions of the images shown in (**C**). All of the axons that were found in both area (a) and (b) and that could be traced for >1,000 µm were analyzed (14 axons). (**E**) Ternary plots of color codes in the axons highlighted in (**C**). Average color codes in each area (a–c) are shown. The color consistency is shown on the right. The color code was largely consistent across the areas except for a few axons (#6, #11 or #12). The copy number may not be optimum in this sample, as the plots are not found in the periphery of the ternary plots. The fluorescence intensity data used for (**E**) are available in *Figure 8—source data 1*. (**F**) Tracing and reconstruction of M/T cell axons. We terminated tracing when the labeled axons were interrupted by unlabeled gaps. Thus, our conservative tracing is most probably showing an underestimate of the entire wiring diagram. All of the axons that could be successfully traced for >1,000 µm are shown. The maximum length of the traced axon was 6852.63 µm. A z-stacked image of the reconstruction is shown. Tracings of individual M/T cells are shown in *Figure 8—figure supplement 2*. See also *Video 6*. Scale bars are 1 mm.

DOI: https://doi.org/10.7554/eLife.40350.030

The following source data and figure supplements are available for figure 8:

**Source data 1.** Fluorescence intensity data used for *Figure 8*E.

DOI: https://doi.org/10.7554/eLife.40350.034

**Source data 2.** Axon length data used for *Figure 8—figure supplement 1D*.

DOI: https://doi.org/10.7554/eLife.40350.033

**Figure supplement 1.** Comparison of the tracing performances of single-color and Tetbow labeling using dense axon labeling.

DOI: https://doi.org/10.7554/eLife.40350.031

**Figure supplement 2.** Traced M/T cell axons in *Figure 8F* shown separately.

DOI: https://doi.org/10.7554/eLife.40350.032

plasmids are shown in figure supplements. The Tetbow plasmids and their sequences have been deposited at Addgene (https://www.addgene.org/Takeshi_Imai/) with Addgene #104102–104112. See also **SeeDB Resources** (https://sites.google.com/site/seedbresources/) for updated information.

As for the Brainbow experiments, we used AAV-EF1a-BbTagBY (Addgene #45185, RRID: Addgene_45185) and AAV-EF1a-BbChT (Addgene #45186, RRID: Addgene_45186) plasmids.

## Mice

All animal experiments were approved by the Institutional Animal Care and Use Committee of the RIKEN Kobe Institute and Kyushu University. ICR mice (Japan SLC, RRID: MGI: 5652524) were used for *in utero* electroporation and C57BL/6N mice (Japan SLC, RRID: MGI: 5658686) were used for AAV experiments (age, P56-70; male). To obtain brain tissue, mice were i.p. injected with an overdose of nembutal (Dainippon Sumitomo Pharma) or somnopentyl (Kyoritsu Seiyaku) to produce deep anesthesia, followed by an intracardiac perfusion with 4% paraformaldehyde (PFA) in phosphate buffered saline (PBS). Excised brain samples were post-fixed with 4% PFA in PBS at 4°C overnight. Samples were then embedded in 4% agarose and cut into slices of 220, 500, or 1000 µm thick with a microslicer, PRO7N (Dosaka EM).

## *In utero* electroporation

The *in utero* electroporation of plasmids to the cerebral cortex and mitral cells was performed as described previously (*Saito, 2006*; *Ke et al., 2013*; *Muroyama et al., 2016*). Pregnant ICR mice were anesthetized with medetomidine (0.3 mg/kg), midazolam (4 mg/kg), and butorphanol (5 mg/kg), or with ketamine (100 mg/kg) and xylazine (10 mg/kg). The uterine horns carrying embryos were exposed through a midline abdominal incision. To label L2/3 neurons in the cortex, 1 µL of plasmid solutions diluted in PBS was injected into the lateral ventricle of the embryos at E15 using a micropipette made from a glass capillary. Electric pulses (single 10 ms poration pulse at 72 V, followed by five 50 ms driving pulses at 40–42 V with 950 ms intervals) were delivered by a CUY21EX electroporator (BEX) and forcep-type electrodes (5 mm diameter, #LF650P5, BEX). To introduce pCAG-XFP vectors, the labels pCAG-mTurquoise2, EYFP, and tdTomato were injected into the lateral ventricle and electroporated at high-copy (1 µg/µL each, 3 µg/µL in total) or low-copy (0.25 µg/µL each, 0.75 µg/µL in total) numbers. pTRE-XFP vectors with pCAG-tTA were injected and electroporated at low-

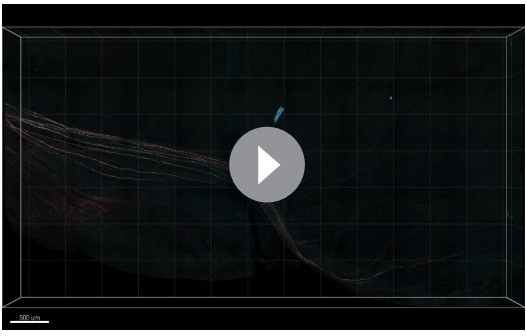

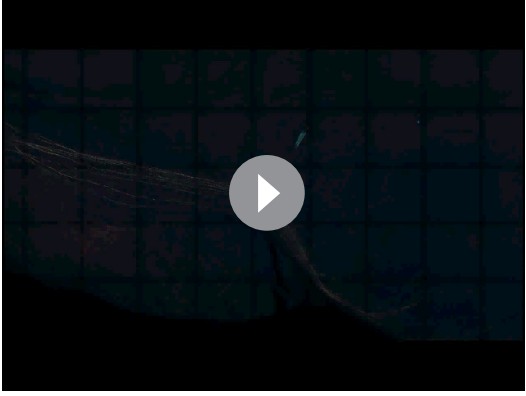

**Video 5.** M/T cell axons sparsely labeled with Tetbow AAVs. Tetbow AAVs were injected to the olfactory bulb at P60 and analyzed after four weeks. Volume-rendering images of the olfactory cortex are shown. See legends to *Figure 8*.
DOI: https://doi.org/10.7554/eLife.40350.035

**Video 6.** Reconstruction of M/T cells axons. Traced M/T cell axons are separately shown. See legends to *Figure 8* and *Figure 8—figure supplement 1*.
DOI: https://doi.org/10.7554/eLife.40350.036

copy numbers (0.25 µg/µL each, 1.0 µg/µL in total) when not specified; in *Figure 3*, the exact plasmid concentrations are specified. To label mitral cells in the olfactory bulb, *in utero* electroporation was performed at E12. Electric pulses (single 10 ms poration pulse at 72 V, followed by five 50 ms driving pulses at 36 V with 950 ms intervals) were delivered with forceps-type electrodes (3 mm diameter, #LF650P3, BEX). After the electroporation, the uterine horns were placed back into the abdominal cavity, and the abdominal wall and skin were sutured.

## AAV

AAV vectors were generated using the AAVpro Helper Free System (AAV1, #6673, Takara) from Takara and the AAVpro 293 T cell line (#632273, Clontech) following the manufacturers' instructions. The backbone pAAV plasmid is for AAV2. Thus, the serotype used in this study is AAV2/1. AAV vectors were generated by AAVpro 293 T cells (#632273, Clontech). AAVpro 293T is a commercialized cell line for production of AAV vectors. We did not test for mycoplasma contamination in our hands. In our experiments, cells within 10 passages were used for virus production. AAV vectors were purified using the AAVpro Purification Kit All Serotypes (#6666, Takara). Virus titers were then determined by qPCR using the AAVpro Titration Kit (#6233, Takara) and the StepOnePlus system (ThermoFisher). To infect the AAV vectors, C57BL/6 mice (P56-70) were anesthetized with ketamine (100 mg/kg) and xylazine (10 mg/kg), and an AAV virus cocktail was injected into the brain using the nanoject II system and glass capillaries (#3-00-203-G/XL, Drummond). The injection volume was 207 nL in *Figure 7*, 138 nL in *Figures 8*, and 207 nL ×5 different locations in *Figure 8—figure supplement 1*. The final concentration of the virus cocktail was $4 \times 10^8$ gc/ml for AAV2/1-*SYN1*-tTA2 and $1 \times 10^{10}$–$1 \times 10^{11}$ gc/mL each for AAV2/1-TRE-mTurquoise2-WPRE, AAV2/1-TRE-EYFP-WPRE, and AAV2/1-TRE-tdTomato-WPRE. The mice were sacrificed 2 or 4 weeks after viral injection. It should be noted that the expression of Tetbow AAV can lead to toxic effects for cellular functions after prolonged incubation because of the extremely high levels of expression. For example, cortical neurons started to show aggregation of XFPs and morphological abnormality 4 weeks after virus injection. Olfactory bulb neurons were best visualized 4 weeks after virus injection without obvious sign of toxicity. The optimum timing for the analysis may be different for different cell types.

The following stereotaxic coordinates were used for AAV injection. Distance in millimeters from the Bregma for the anterior (A) – posterior (P), and lateral (L) positions, and from the brain surface toward the ventral (V) directions are indicated. Cortical layer five neurons: P=1.5, L = 1.5, V = 0.5; hippocampal CA1 neurons: P=1.5, L = 1.5, V = 1; olfactory bulb granule cells: A = 4.5, L = 0.5, V = 0.5; olfactory bulb M/T cells: A = 4.5, L = 0.5, V = 0.3.

## Staining chemical tags

SNAP, CLIP, and HaloTags were visualized with their substrates, SNAP-Surface 488 (#S9124S, New England Biolabs), HaloTag TMR Ligand (#G8252, Promega), CLIP-Surface 647 (#S9234S, New England Biolabs), respectively. Brain slices of 220 or 1000 µm thickness were incubated with the substrates (2 µM each) in 2 ml 2% saponin in PBS overnight. The slices were then washed with PBS (3 × 30 min).

## Clearing with SeeDB2G

Brain-slice samples were cleared with SeeDB2G for imaging when not specified. SeeDB2G is designed for high-resolution imaging with glycerol-immersion objective lenses (*Ke et al., 2016*). PFA-fixed brain samples (embedded in agarose, cut at 220, 500, or 1000 µm thick) were cleared at room temperature (25°C) with a 1:2 mixture of Omnipaque 350 (#081–106974, Daiichi-Sankyo) and $H_2O$ with 2% saponin (#30502–42, Nakalai-tesque) for 6 hr (3 ml in 5 ml tube), a 1:1 mixture of Omnipaque 350 and $H_2O$ with 2% saponin for 6 hr, and finally Omnipaque 350 with 2% saponin overnight. Cleared samples were then mounted in SeeDB2G (Omnipaque 350) on a glass slide using a 0.2 mm thick silicone rubber sheet (AS ONE, #6-9085-13, Togawa rubber) and glass coverslips (#0109030091, Marienfeld, No. 1.5H) (*Ke et al., 2016*). A detailed step-by-step protocol has been published in bio-protocol (*Ke and Imai, 2018*).

To quantify the fluorescence intensity of M/T cell axons (*Figure 2—figure supplement 2*), whole-brain samples were cleared with SeeDB2 and placed on a glass-bottomed dish for imaging and quantification.

To analyze long-rage axonal projection of M/T cells (*Figure 8*), a right-brain hemisphere was dissected, and the dorsal part and subcortical matter was trimmed away with forceps and a scalpel. The remaining part, containing all of the olfactory cortical areas, was flattened with a 700 mm spacer and fixed with 4% PFA in PBS overnight (*Sosulski et al., 2011*). Then, the sample was treated with ScaleCUBIC-1 (25% (wt/wt) urea (#219–00175, Wako), 25% (wt/wt) N,N,N',N'-tetrakis(2-hydroxypropyl)ethylenediamine (#T0781, TCI), and 15% (wt/wt) Triton X-100 (#12967–45, Nakalai-tesque) in $H_2O$) (*Susaki et al., 2014*) for 24 hr to remove lipids from the lateral olfactory tract, washed with PBS, and then cleared with SeeDB2G as described above.

## BABB

PFA-fixed brain samples were serially incubated in 50%, 80%, and 100% ethanol, each for 8 hr. They were then incubated in 100% ethanol for 12 hr, and then in hexane (#17922–65, Nakalai-tesque) for 12 hr. Samples were cleared (benzyl alcohol (# 402834–100 ML, SIGMA):benzyl benzoate (#025–01336, Wako)=1:2) with gentle shaking for 24 hr.

## 3DISCO

PFA-fixed brain samples were serially incubated in 50%, 70%, 80%, and 100% tetrahydrofuran (THF, # 207–17905, Wako), each for 1 hr. They were then incubated in 100% THF for 12 hr, and then in dibenzyl ether (DBE, #022–01466, Wako) for 3 hr (*Ertürk et al., 2012*).

## Confocal imaging

Confocal images were acquired using an inverted confocal microscope, TCS SP8X with HyD detectors (Leica Microsystems). Type G immersion (refractive index 1.46, Leica) was used for a 20x multi-immersion objective lens (HC PL APO 20x/0.75 IMM CORR CS2, NA0.75, WD 0.66 mm) and a 63x glycerol-immersion objective lens (HC PL APO 63x GLYC CORR CS2, NA1.3, WD 0.28 mm). Type F immersion (refractive index 1.518, Leica) was used for a 40x oil-immersion objective lens (HC PL APO 40x OIL CS2, NA1.3, WD 0.24 mm). Low-magnification images in *Figure 2—figure supplement 2* were taken with a 10x dry lens (HC PL APO 10x/0.40 CS, NA0.4, WD 2.2 mm). Diode lasers of 442 or 448 nm, 488 nm, and 552 nm wavelength were used for mTurquoise2, EYFP, and tdTomato, respectively. In some experiments, a white light laser was used to image mTurquoise2, EYFP, and tdTomato. To image chemical tags, SNAP-Surface 488, HaloTag TMR Ligand, and CLIP-Surface 647 were imaged with diode lasers of 488 nm, 552 nm, and 638 nm wavelength, respectively. Emission light was dispersed by a prism and detected by HyD detectors.

## Image processing and quantification

Microscopy data were processed and visualized with LAS X (RRID: SCR_013673, Leica Microsystems) or Imaris (RRID: SCR_007370, Bitplane). Image data were excluded from further quantitative analyses when the data contained saturated fluorescence signals. For *Figures 2* and *3*, the fluorescence intensities on somata were quantified with ImageJ at every 2.97 µm thickness. For *Figure 7*, an image of the soma was taken and analyzed at the focal depth for each cell. For *Figure 8*, color values at axons were analyzed for maximum intensity projection images, as there may be a slight chromatic aberration along the z axis. After binarization, the average color value ratio was determined for each segment of each axon. Ternary plots, box plots, and statistical analyses were prepared using MATLAB (RRID: SCR_001622, MathWorks). Image data were acquired by RS, to prevent bias when they were subsequently analyzed by MNL. First, the fluorescence intensity in each channel was normalized so that the median intensity was 1. Then, the intensity values were further normalized so that the length of the vector was 1. Raw quantification data (Source Data files) are accompanied with figures. The MATLAB code and processed data for the figure panels have been deposited to GitHub (*Leiwe, 2018*; copy archived at https://github.com/elifesciences-publications/TetbowCodes).

## Axon tracing

Axon tracing was performed with Neurolucida (RRID: SCR_001775, MBF Bioscience). In *Figure 8F*, we focused on axons were found within 200 µm of the anterior end of the image. We terminated tracing when labeled axons were interrupted by unlabeled gaps; in such cases, we judged that we cannot be 100% sure whether the interrupted segments are connected. Thus, our tracing was performed using very conservative criteria, and thus most probably provides an underestimate of the entire wiring diagram. Axons that can be traced for >1,000 µm were analyzed (25 axons). Putative mitral and tufted cells were identified on the basis of the projection area (piriform cortex vs. olfactory tubercle) and their axonal trajectories. Brightly-labeled putative mitral and tufted cells were further analyzed in *Figure 7D,E*. For *Figure 8—figure supplement 2*, we focused on axons that crossed the anterior plane and within the lateral half. We chose 100 axons in an unbiased manner from the dorsal to ventral direction. The criteria for successful tracing are described in *Figure 8D*; as these criteria are very strict, the area that is traced may be an underestimate. Color discrimination was performed visually by one experimenter, who also joined the visual color discrimination test.

## Visual color discrimination

Custom MATLAB code (*Leiwe, 2018*; copy archived at https://github.com/elifesciences-publications/TetbowCodes) was written to quantify the ability of experienced researchers to discriminate three color space (RGB) in terms of 3D Euclidean space. All of the subjects (from the Imai lab) have experience in fluorescence imaging, understand the purpose of the test, and have trichromatic color vision. To prevent the intensity of the color from influencing the decision, the intensity was randomly varied between 50–100% for each square displayed (*Figure 1—figure supplement 1*). The subject was presented with a choice to discriminate between two colors at a specified Euclidean distance. Specifically, they were asked to determine whether the two columns form a cross or a parallel line. For each Euclidean distance presented, there were 100 trials per subject, with the average success rate stored. Note that this test was not intended to evaluate the color discrimination performance of people in general.

## Modeling

Poisson distributions were calculated in MATLAB, by creating independent distributions for each XFP for each specified copy number. 200 'cells' were selected for each copy number group with associated XFP values derived from the Poisson distributions. The code has been deposited to GitHub (*Leiwe, 2018*; copy archived at https://github.com/elifesciences-publications/TetbowCodes).

## Imaging data

The raw microscopy data have been deposited to the Systems Science of Biological Dynamics (SSBD) database (http://ssbd.qbic.riken.jp/) with a unique URL (http://ssbd.qbic.riken.jp/set/20180901/). Movies will be posted at SeeDB Resources (https://sites.google.com/site/seedbresources/).

## Step-by-step protocol

Step-by-step protocols has been posted in our website, **SeeDB Resources** (https://sites.google.com/site/seedbresources/). All of our published clearing methods and technical tips are also posted on this website. A detailed protocol for SeeDB2 was published in bio-protocol (*Ke and Imai, 2018*).

## Acknowledgements

This work was supported by the program for Brain Mapping by Integrated Neurotechnologies for Disease Studies (Brain/MINDS) from Japan Agency for Medical Research and Development (AMED) (JP18dm0207055h to TI), a grant from the programs Grants-in-Aid for Scientific Research on Innovative Areas 'Dynamic regulation of Brain Function by Scrap & Build System' (JP16H06456 to TI) from MEXT, JSPS KAKENHI (JP17H06261, JP16K14568, JP15H05572, and JP15K14336 to TI; JP17K14946 to MNL), Brain Science Foundation (to TI), and RIKEN CDB intramural grant (to TI). RS was a Junior Research Associate at RIKEN. Imaging experiments were supported by the RIKEN Kobe Light Microscopy Facility. Animal experiments were supported by LARGE. We also appreciate technical assistance from The Research Support Center, Research Center for Human Disease Modeling, Kyushu University.

## Additional information

### Funding

| Funder | Grant reference number | Author |
| --- | --- | --- |
| Japan Society for the Promotion of Science | JP17K14946 | Marcus N Leiwe |
| Japan Agency for Medical Research and Development | JP18dm0207055h | Takeshi Imai |
| Japan Society for the Promotion of Science | JP17H06261 | Takeshi Imai |
| Brain Science Foundation | | Takeshi Imai |
| RIKEN | | Takeshi Imai |
| Japan Society for the Promotion of Science | JP16H06456 | Takeshi Imai |
| Japan Society for the Promotion of Science | JP16K14568 | Takeshi Imai |
| Japan Society for the Promotion of Science | JP15H05572 | Takeshi Imai |
| Japan Society for the Promotion of Science | JP15K14336 | Takeshi Imai |

The funders had no role in study design, data collection and interpretation, or the decision to submit the work for publication.

### Author contributions

Richi Sakaguchi, Data curation, Formal analysis, Validation, Investigation, Visualization, Methodology, Writing—original draft, Writing—review and editing, Performed most of the wet experiments, Analyzed data; Marcus N Leiwe, Software, Formal analysis, Supervision, Validation, Investigation, Visualization, Writing—original draft, Writing—review and editing, Performed stimulations and color discrimination analysis, Analyzed data; Takeshi Imai, Conceptualization, Supervision, Funding acquisition, Methodology, Writing—original draft, Project administration, Writing—review and editing, Conceived the experiments, Supervised the project

Author ORCIDs
Richi Sakaguchi http://orcid.org/0000-0002-3988-6385
Marcus N Leiwe http://orcid.org/0000-0002-3493-7091
Takeshi Imai https://orcid.org/0000-0002-8449-0080

## Ethics

Animal experimentation: All animal experiments were approved by the Institutional Animal Care and Use Committee of the RIKEN Kobe Institute (#AH-02-23) and Kyushu University (#A29-241-1).

## Decision letter and Author response

Decision letter https://doi.org/10.7554/eLife.40350.041
Author response https://doi.org/10.7554/eLife.40350.042

## Additional files

### Supplementary files

• Transparent reporting form
DOI: https://doi.org/10.7554/eLife.40350.037

### Data availability

Raw microscopy data have been deposited to Systems Science of Biological Dynamics (SSBD) database (http://ssbd.qbic.riken.jp/) with a unique URL (http://ssbd.qbic.riken.jp/set/20180901/). MATLAB code and processed data for figure panels have been deposited to GitHub (https://github.com/mleiwe/TetbowCodes; copy archived at https://github.com/elifesciences-publications/TetbowCodes). A Source data file includes numerical data.

The following dataset was generated:

| Author(s) | Year | Dataset title | Dataset URL | Database and Identifier |
|---|---|---|---|---|
| Richi Sakaguchi, Marcus N Leiwe, Takeshi Imai | 2018 | A set of imaging data of neuronal circuits labeling with fluorescent proteins in M. musculus | http://ssbd.qbic.riken.jp/set/20180901/ | Systems Science of Biological Dynamics, 20180901 |

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
