## [Decision Letter]

[Editors’ note: a previous version of this study was rejected after peer review, but the authors submitted for reconsideration. The first decision letter after peer review is shown below.]

Thank you for submitting your work entitled "Tetbow: bright multicolor labeling of neuronal circuits with fluorescent proteins and chemical tags" for consideration by *eLife*. Your article has been reviewed by three peer reviewers, including Moritz Helmstaedter as the Reviewing Editor and Reviewer #1, and the evaluation has been overseen by a Reviewing Editor and a Senior Editor.

Our decision has been reached after consultation between the reviewers.

Based on these discussions and the individual reviews below, we regret to inform you that your work will not be considered for publication in *eLife* at this point. However, if you are able to provide the additional experiments outlined below, we would be willing to consider a revised manuscript. This decision reflects the fact that *eLife* does not allow reviewers to suggest new experiments that are likely to take longer than 2 months to perform for a revision decision. By instead rejecting these manuscripts, we leave it up to the authors to decide if they think the reviewers are essentially correct, in which case, they may wish to do the requested experiments. If, however, they do not agree with the reviewers, then we would expect authors to send the paper elsewhere. Of course, if you already have the requested experiments in hand, we would be very pleased to have a revised manuscript, incorporating those new data immediately.

Your manuscript reports an approach for bright multicolor labeling of neurons with the goal of deducing neuronal circuitry. While the reviewers see a benefit of this technology over existing techniques, the following points would need to be added to the manuscript to satisfy the consensus concerns of the reviewers (see below for unedited original reviews):

1) While it may seem plausible that an improvement of color distinction and/or brightness of labeling and/or constancy of colors were achieved, a clear application demonstrating the advance over existing techniques is missing. The reviewers suggest either of the following two proof-of-principle reports:

- Quantitative report of long-distance (millimeters) color constancy when labeling axons.

- Automated / efficient reconstruction of single neurons over longer distances, quantification that this reconstruction is performing better with TetBow than with previous techniques.

2) The implications of the technique for circuit reconstruction (connectomics) should be described more cautiously: in local dense networks, the fluorescent labeling of neurons alone does not allow the inference of synaptic connectivity. For long-range labeling, the situation is more benign. We would ask that you amend the text accordingly and discuss this, also in light of the requested proof-of-concept application (see point 1).

Without a proof-of-principle experiment showing clearly the advance of the method over existing technology, the evaluation of the manuscript would be substantially less positive.

Reviewer #1:

The manuscript "Tetbow: bright mutlicolor labeling of neuronal circuits with fluorescent proteins and chemical tags" by Sakaguchi et al. reports the development of a multicolor neuronal labelling toolbox that uses the Tetracyclin-operator system instead of the Cre-loxP system used for the "Brainbow" technology. The main claim of the manuscript is that this different approach increases color variability and provides enhanced color intensity. While an improvement over the "Brainbow" technology for neuron labelling is a relevant endeavour.

I see the following key concerns with the manuscript:

1) The notion of wiring diagram and circuits is mistaken when all that is provided is an intracellular labelling of a presynaptic neuronal population. Wiring diagrams imply synapse detection which this method is currently not providing. This has to be made very clear and care should be taken not to confuse the terminology. Still, intense and long-range labelling of a large number of presynaptic neurons can contribute to circuit analysis and in this sense the method could be valuable.

2) While the manuscript contains many beautiful images, a quantitative documentation of long-range color constancy is missing. In this reviewer's opinion the long-range color constancy is a key prerequisite to use multicolor labelling methods at the light level for circuit inference. While locally the detection of synapses is absolutely required to distinguish an incidental from a synaptic contact, millimeters away from the source neuron, circuit inference can be plausibly done by judging the projection target regions of presynaptic neurons. However, for this the neurons have to stay brightly labelled over millimeters if not centimeters in larger brains. This needs to be documented and quantified for this approach to make a substantial advance.

3) Related to point 2, it would be necessary to show a clear, at least potential improvement in terms of interpretation of such data. Some proof of concept application would be required – at least a hypothetical one. This does not imply that for a methods manuscript a full result has to be documented but at least the notion of what this kind of result could be and why this method in contrast to previous methods will be able to achieve this. Again, I would suggest using the long-range circuit inference as one of these possible key applications.

In summary I think this is an interesting methodological advance that however so far lacks clear quantitative documentation of long-range color constancy. The enhancement of color space is impressive but without a clear application difficult to judge in terms of its impact for neuroscience.

Reviewer #2:

In this article the authors present a useful variation of Brainbow strategy for stochastic multicolor labeling of neurons. Their method, which they called Tetbow, combines the Tet-Off system with the Brainbow approach. They show that they can generate multicolor labeling using plasmids for XFPs or chemical tags as well as with viral tools. These reagents will be a useful addition to the already available recombinase based or transgenic toolkit for multicolor labeling. However, the authors' major claim that the expression levels of FPs using this approach is much higher, therefore this strategy 'should facilitate neuronal circuit reconstruction at higher densities and resolutions' compared to the current -best of class- Brainbow approaches is not supported by the data presented. Labeling many neurons in a brain and labeling individual neurons bright enough for complete reconstructions is a challenging problem. But it is not clear from the data presented that the approach presented in this manuscript solves this problem.

Specific comments:

1) The conditions for induction of the Tet-Off system need to be described in the Materials and methods section.

2) It is not clear from the description why the 0.25µg/µl was chosen for the in utero electroporations. Were multiple concentrations tried? Was the chosen concentration arrived at after examining spread (colors) or by looking at ternary plots as in Figure 2D.

3) Subsection “Image processing and quantification”, subsection “Modeling”, subsection “Imaging data”, include code and data location.

4) Figure 2 panel D – the ternary plot could be separated out for the three conditions for clarity.

5) Figure 1—figure supplement 1 seems unnecessary – addresses a special case of Brainbow.

6) Figure 2—figure supplement 1 and Figure 2—figure supplement 2 can be condensed together; the plasmid cartoons again could be condensed. Similarly, for the other plasmid and vector maps.

7) Figure 2—figure supplement 3 – Brainbow 3.0 experiment is uninformative.

Reviewer #3:

In this manuscript, Sakaguchi et al., present a toolbox for multicolor neuronal labeling termed "Tetbow". Their approach relies on mixing three distinct vectors that express different colors of fluorescent protein (cyan, yellow or red) with the Tet-Off system. Each fluorescent protein gene is under the control of a tetracycline response element (TRE), activated by a transactivator (tTA2) encoded by a fourth vector. The authors show that this strategy enables multicolor labeling of mouse neurons by in utero electroporation, and provide evidence that higher expression and more color contrast can be achieved with Tet-Off transactivation compared with direct expression of the fluorescent proteins from a CAG promoter. They also present a variation of their technique in which protein tags (SNAP, Halo and CLIP) are used to label neurons with combinations of synthetic fluorochromes resistant to tissue clearing procedures. Finally, they present a version of Tetbow based on AAV vectors, which also achieves multicolor labeling of neurons in injected brain areas.

The tools presented in the manuscript may be of interest for the neuroscience community and several convincing images are provided that support their effectiveness. I have however several general concerns about the manuscript:

First, most of the concepts used in the paper are not new. For instance, multicolor labeling with mixtures of vectors expressing distinct XFPs has been introduced several years ago (Weber et al., 2011), as has been the usage of AAV vectors (Cai et al., 2013) or electroporation (Loulier et al., 2014) to achieve multicolor neuronal labeling, and the modeling of the relation between copy number and color combinations (Kobiler et al., 2010).

Second, beyond the images presented in the article, there is no demonstration of a usage of the Tetbow approach to trace connectivity. An application of these strategies to study some aspects of brain circuitry is essential to evaluate their usefulness. In particular it appears uncertain that labeled neuronal processes can be followed in their entirety in neural tissue samples.

Third, due to incomplete description or inappropriate evaluation procedures, the actual improvement brought by Tetbow appears uncertain. For instance, it is unclear if the number of samples analyzed in Figure 1 is sufficient to minimize discrepancies among the different brains analyzed, and between different sections of a given brain. The authors also claim that their scheme is simpler that Brainbow, but in practice this is hardly the case: with Tetbow, a total of 4 plasmids or AAV vectors must be introduced in neurons of interest, while with Brainbow two plasmids (Brainbow et al., 2014) or 2 AAVs (AAV no. 1 and no. 2, Cai et al., 2013) are sufficient. In addition, with Tetbow, careful titration of the different color vectors needs to be performed for each fresh vector preparation, while expression of different XFPs is intrinsically balanced in Brainbow transgenes, making them inherently more reproducible. Concerning the strategy based on protein tags (Figure 5), one cannot assess how deep synthetic fluorochromes labels diffuse within tissue sections: does this scheme really enable to label more than just the first few 10th of micrometers of tissue samples? Finally, are Tetbow protein tags (Figure 5) and AAVs (Figure 6) more efficient in terms of expression level and color contrast than standard vectors? It seems that most cells in Figure 5 and Figure 6 coexpress all three XFPs and that the color contrast in these samples is low.

[Editors’ note: what now follows is the decision letter after the authors submitted for further consideration.]

Thank you for resubmitting your work entitled "Tetbow: bright multicolor labeling of neuronal circuits with fluorescent proteins and chemical tags" for further consideration at *eLife*. Your revised article has been favorably evaluated by Eve Marder (Senior Editor), a Reviewing Editor, and three reviewers.

The manuscript has been improved but there are some remaining issues that need to be addressed before acceptance, as outlined below:

While the reviewers were very positive about the extensive revision, they concluded after extended discussion that they would strongly recommend to address the following remaining issues. These will likely not require an additional consultation with the reviewers after submission of the final revision.

- The text needs substantial revision along the suggestions of reviewers 2 and 3, especially to properly reference the literature and remove ambiguous statements.

- We recommend describing the quality of terminal axonal branch labeling, since this is considered a key benchmark of this set of methods.

Reviewer #1:

This manuscript is a revision of an earlier submission. The authors have taken serious steps to address the issues raised before and have in my view substantially improved the manuscript. The case for Tetbow being a significant step forward for multi-color neuronal labeling has been made more clearly.

In particular, I think the added analyses of color assessment, comparison of fluorescence intensity to Brainbow; and importantly the analysis of color constancy over long distances (Figure 8 and associated supplements) are valuable and relevant.

*Reviewer #2:*

The revised manuscript, Tetbow: bright mutlicolor labeling of neuronal circuits with fluorescent proteins and chemical tags" by Sakaguchi et al., addresses some of the points raised by the reviewers in the previous submission however, one significant concern remains.

This methods paper adds to the existing toolkit for multicolor labeling of neurons. The tTA-*TRE* based reagents described in this manuscript appear to have improved brightness over the traditional Brainbow methods. While such a color palette would certainly be useful for answering certain neuroanatomical questions the revised manuscript still fails to address the primary concern that was raised in the original submission – whether such a multicolor labeling approach would be useful for complete neuronal reconstructions. Will the use of multicolor labeling permit reconstruction of entire neurons at higher densities as the authors suggest, i.e. is it an improvement over existing methods that simply use sparse and bright neuronal labeling with a limited color-set.

Figure 8 and Figure 8—figure supplement 2 in this revised submission address this question. The data presented clearly show that the trajectories of the main axon of multiple M/T neurons can be traced. This might be useful if the goal was to identify the primary brain areas targeted by these neurons. But it is not at all clear from the data shown whether complete reconstructions of neurons would be feasible and therefore prompts the question if Tetbow labeling is bright enough to trace axonal arbors in entirety. Reconstructions that aim to trace out only the main axon and the first order collaterals are already possible even in Thy1-GFP transgenic animals (see for instance Guo et al., (2017)). The authors would have definitely compared their traced neurons to the Mitral/Tufted cell reconstructions presented in Igarashi, 2012 or Ghosh, 2011 (both articles are cited in the manuscript). Is it possible to get similar level of completeness using the Tetbow approach?

Reviewer #3:

Sakaguchi et al., present a revised version of their article on the development of an enhanced multicolor neuronal labeling toolbox termed "Tetbow". They have significantly strengthened their study, with the addition of: (1) a theoretical analysis of the color discriminability expected with their labeling scheme; (2) a more precise characterization of the effects of plasmid and AAV concentrations and ratios on color labels; (3) examples of axon tracing over long distances in mitral cells labeled with AAV mixtures. This latter point constituted the main insufficiency of the initial draft of the article, and is addressed in a relatively convincing manner, although an evaluation of the applicability of chemical tags for tracing is still lacking. The tools presented in the manuscript and the efforts made at characterizing optimal labeling conditions will be of interest for the neuroscience community.

My main comment about this new version of the article concerns the presentation of the results, which requires significant revision of the text for the following reasons:

- Several concepts presented in the article are not novel per se and are simply improved and more deeply explored than in the studies that introduced them. The true interest of the article lies in these improvements and optimizations, which will undoubtedly be useful to the community, not in the rediscovery of previous ideas. This is for instance the case of multicolor labeling with mixtures of distinct single-color viral vectors (Chan et al., 2017; Weber et al., 2011) or plasmids (Siddiqi et al., 2014), modeling of the relationship between copy number and color (Kobiler et al., 2010), usage of the Tet-Off system to amplify expression in a multicolor context (Chan et al., 2017 and other studies)…When introducing these concepts, the text should state this explicitly and refer to the related articles. It is with respect to these recent studies that the new tools should be judged, not only relative to former Cre-dependent Brainbow approaches.

- The discussion should also include a section on the drawbacks of Tetbow and possible difficulties of this approach, such as the necessity to carefully titrate the different RGB vectors to equilibrate their concentration, batch-to-batch variations, and whether it is compatible with strategies to sparsen expression e.g. as in Chan et al., 2017.

-Some statements are repeated unnecessarily throughout the paper (e.g. the known fact that Cre recombination is not required for combinatorial labeling).

- Several sentences are quite imprecise, some conclusions are overstated and there are also many typos that need to be corrected.

---

## [Author Response]

[Editors’ note: the author responses to the first round of peer review follow.]

[…] 1) While it may seem plausible that an improvement of color distinction and/or brightness of labeling and/or constancy of colors were achieved, a clear application demonstrating the advance over existing techniques is missing. The reviewers suggest either of the following two proof-of-principle reports:- Quantitative report of long-distance (millimeters) color constancy when labeling axons.- Automated / efficient reconstruction of single neurons over longer distances, quantification that this reconstruction is performing better with TetBow than with previous techniques.

We agree that it is important to demonstrate the capabilities of our bright multicolor-labelling method, Tetbow. Particularly, proof-of-principle experiments such as tracing long-range axonal projections would be helpful. In our new sets of experiments, we focused on mitral and tufted (M/T) cells in the olfactory bulb (OB), which project axons (up to ~6 mm long) to the olfactory cortices, including the olfactory tubercle, piriform cortex, cortical amygdala, and lateral entorhinal cortex. When we labeled M/T axons with Tetbow, axon collaterals of M/T cells were clearly visualized in the piriform cortex; however, published Brainbow constructs could not clearly label these axons in the same condition (Figure 2—figure supplement 2).

Automated circuit tracing is obviously an important future direction using Tetbow; however, it is far beyond the scope of the current submission to develop and include auto-tracing software. Instead, we performed manual tracing. To evaluate the performance of our manual tracing objectively, we examined the color discrimination performance of researchers (Figure 1—figure supplement 2). We confirmed that we can visually discriminate color hues with 0.1 Euclidian distance in a C-Y-R color coding space with >90% accuracy (Figure 1H). In that situation, one cell can be successfully discriminated from ~90% of random cells (Figure 1G).

We then evaluated the performance of Tetbow for manually tracing M/T cell axons. In one experiment (Figure 8), we injected AAV-Tetbow locally. In another experiment (Figure 8—figure supplement 2), we injected the same virus cocktail to the entire dorsal OB densely. Tracing was performed after tissue clearing. In the local injection experiments, only dozens of M/T cells were labelled, and these axons were clearly distinguishable from each other and could be traced into the olfactory cortex (Figure 8). Each M/T cell demonstrated a unique branching pattern in the cortex (Figure 8—figure supplement 1), supplementing previous studies that used laborious single-axon tracing (Nagayama et al., 2010; Sosulski et al., 2011; Ghosh et al., 2011; Igarashi et al., 2012). Notably, the unique color hue was maintained from proximal to distal part of M/T cell axons, allowing for reliable manual tracing (Figure 8C). In the second experiment (dense labeling), most of M/T cells in the dorsal OB were labelled (Figure 8—figure supplement 2A); however, these axons were dissociated and traced for hundreds of microns based on color hues, which was impossible with single-color labeling (Figure 8—figure supplement 2B, C). Together, we concluded that our Tetbow method allows for long-range axonal tracing (at several millimeters scale) in cleared tissues at unprecedented reliability.

2) The implications of the technique for circuit reconstruction (connectomics) should be described more cautiously: in local dense networks, the fluorescent labeling of neurons alone does not allow the inference of synaptic connectivity. For long-range labeling, the situation is more benign. We would ask that you amend the text accordingly and discuss this, also in light of the requested proof-of-concept application (see point 1).

We agree that our method cannot validate synaptic connectivity, while synaptic structures such as dendritic spines and axon boutons can be clearly visualized. We also understand that some researchers use “connectomics” or “wiring” strictly for synapses, whereas others use them also for mesoscopic and macroscopic levels from an area to another area. In the new experiments for tracing long-range projections, we merely analyzed the branching patterns of individual M/T cells. To avoid unnecessary confusions, we amended the text accordingly. We have changed “connectomics” to “neuronal tracing”.

Reviewer #1:[…] 1) The notion of wiring diagram and circuits is mistaken when all that is provided is an intracellular labelling of a presynaptic neuronal population. Wiring diagrams imply synapse detection which this method is currently not providing. This has to be made very clear and care should be taken not to confuse the terminology. Still, intense and long-range labelling of a large number of presynaptic neurons can contribute to circuit analysis and in this sense the method could be valuable.

We thank this reviewer for recognizing utility of our new method. To avoid any possible confusions, we amended the text, as explained above.

2) While the manuscript contains many beautiful images, a quantitative documentation of long-range color constancy is missing. In this reviewer's opinion the long-range color constancy is a key prerequisite to use multicolor labelling methods at the light level for circuit inference. While locally the detection of synapses is absolutely required to distinguish an incidental from a synaptic contact, millimeters away from the source neuron, circuit inference can be plausibly done by judging the projection target regions of presynaptic neurons. However, for this the neurons have to stay brightly labelled over millimeters if not centimeters in larger brains. This needs to be documented and quantified for this approach to make a substantial advance.3) Related to point 2, it would be necessary to show a clear, at least potential improvement in terms of interpretation of such data. Some proof of concept application would be required – at least a hypothetical one. This does not imply that for a methods manuscript a full result has to be documented but at least the notion of what this kind of result could be and why this method in contrast to previous methods will be able to achieve this. Again, I would suggest using the long-range circuit inference as one of these possible key applications.

We agree and have performed analysis of long-range projections focusing on M/T cells, as detailed above. We also evaluated the reliability of manual (visual) neuronal tracing (Figure 1H).

In summary I think this is an interesting methodological advance that however so far lacks clear quantitative documentation of long-range color constancy. The enhancement of color space is impressive but without a clear application difficult to judge in terms of its impact for neuroscience.

We appreciate the enthusiasm on our manuscript. In the revised manuscript, we added a new analysis of M/T cell projection (Figure 8). The projection patterns of M/T cells from the OB to the olfactory cortex have been a big mystery for many years. While several studies tried axonal tracing from the OB to the olfactory cortex (Nagayama et al., 2010; Sosulski et al., 2011; Ghosh et al., 2011; Igarashi et al., 2012) and partially revealed the logic, single axon tracing has been still laborious. We believe that with our new strategy (a combination of Tetbow labeling and tissue clearing) should open an opportunity for producing and studying a more comprehensive map of long-range axonal projection.

Reviewer #2:[…] These reagents will be a useful addition to the already available recombinase based or transgenic toolkit for multicolor labeling. However, the authors' major claim that the expression levels of FPs using this approach is much higher, therefore this strategy 'should facilitate neuronal circuit reconstruction at higher densities and resolutions' compared to the current -best of class- Brainbow approaches is not supported by the data presented. Labeling many neurons in a brain and labeling individual neurons bright enough for complete reconstructions is a challenging problem. But it is not clear from the data presented that the approach presented in this manuscript solves this problem.

In our previous manuscript, we compared the brightness and color variability using CAG-XFP and Tetbow constructs. We believe that this is a fair comparison, as we used the same amount of plasmids containing the same fluorescent proteins. We, however, understand that it is also important to perform a side-by-side comparison with the current -best of class- Brainbow constructs for realistic samples. To address this issue, we introduced Brainbow AAV plasmids (Cai et al., 2013) and Tetbow plasmids (this study) into M/T cells using *in utero* electroporation. It should be noted that only EYFP is shared between two methods, and the other fluorescent proteins are different.

We analyzed three samples each. As for Tetbow, we could clearly visualize M/T cell axons in the olfactory cortex, millimeters away from the OB. However, we could not recognize any axonal signals in our two Brainbow samples, and we could only detect faintly dotted signals for one sample (Figure 2— figure supplement 2). As we could not recognize axons in 2/3 samples, we cannot evaluate fluorescence intensity quantitatively. However, these results clearly demonstrate the improved neuronal tracing ability of Tetbow. Single axon tracing is also shown in Figure 8. Thus, our method solved the problem.

To demonstrate the power of mulcicolor labeling strategy, we also compared tracing performance for multicolor vs single color (artificially produced from the multicolor data) using densely-labelled challenging sample (Figure 8—figure supplement 1). The result was quite obvious as expected.

Specific comments:1) The conditions for induction of the Tet-Off system need to be described in the Materials and methods section.

The Tet-Off system does not require DOX administration. We simply co-expressed tTA2 with TRE-XFPs. The plasmid amount or virus titer is detailed in the Materials and methods section.

2) It is not clear from the description why the 0.25µg/µl was chosen for the in utero electroporations. Were multiple concentrations tried? Was the chosen concentration arrived at after examining spread (colors) or by looking at ternary plots as in Figure 2D.

We examined four different concentrations of the Tetbow plasmids (0.05, 0.1, 0.25, 0.5 µg/µL). As for the color discriminability, we introduced a new evaluation metric in the revised manuscript. We examined how likely two randomly chosen cells can be discriminated based on color dissimilarity (Euclidian distance in a color coding space). As the discriminability is determined by the threshold Euclidian distance, we examined discriminability at various threshold levels. To place this in context, the accuracy of human vision was also tested. In manual (human vision) recognition, the threshold level was ~0.1 (Figure 1H). In terms of color discriminability, there was no significant difference among the four conditions (Figure 3A-C). To our surprise, the intensity of fluorescent proteins was highest at 0.25µg/µL, not 0.5. We therefore used 0.25µg/µL for the *in utero* electroporation.

We further examined why 0.25 was better than 0.5, and found that a lower concentration of tTA is preferred for higher expression levels of TRE-XFPs (Figure 3D-F). It is possible that an excessive amount of tTA interferes with the transcription machinery. Based on this finding, we also omitted the WPRE sequence for tTA plasmids, and it performed better without WPRE.

3) Subsection “Image processing and quantification”, subsection “Modeling”, subsection “Imaging data”, include code and data location.

All of the codes developed in this study will be uploaded to GitHub. Link to the unique GitHub site will be included in the final manuscript as well as in our website, SeeDB Resources. Similarly, all of the plasmids have been deposited to Addgene (https://www.addgene.org/Takeshi_Imai/). All of the raw image data will be deposited to SSBD database (http://ssbd.qbic.riken.jp/set/20180701/).

4) Figure 2 panel D – the ternary plot could be separated out for the three conditions for clarity.

The ternary plots are now separately shown in Figure 2C, D.

5) Figure 1—figure supplement 1 seems unnecessary – addresses a special case of Brainbow.

As advised, we deleted the old Figure 1—figure supplement 1.

6) Figure 2—figure supplement 1 and Figure 2—figure supplement 2 can be condensed together; the plasmid cartoons again could be condensed. Similarly, for the other plasmid and vector maps.

In our understanding, supplementary figures will not be included in the PDF version, and only seen online linked to each main figure. We therefore think that each figure supplement should be separate in *eLife*. Please advise us if we are wrong.

7) Figure 2—figure supplement 3 – Brainbow 3.0 experiment is uninformative.

We performed the Brainbow3.0 experiment, as this was one of the best Brainbow cassettes published so far (Cai et al., 2013). The conclusion is that fluorescence was too weak for neuronal tracing with Brainbow3.0. However, it is true that Brainbow3.0 expresses mOrange2, EGFP, and mKate, none of which is used in Tetbow. To answer this reviewer’s request, we performed new experiments using another set of Brainbow plasmids, which contain EYFP (Cai et al., 2013). This result is now in new Figure 2—figure supplement 2. Again, the fluorescence was too weak for neuronal tracing in this condition.

Reviewer #3:[…] The tools presented in the manuscript may be of interest for the neuroscience community and several convincing images are provided that support their effectiveness. I have however several general concerns about the manuscript:First, most of the concepts used in the paper are not new. For instance, multicolor labeling with mixtures of vectors expressing distinct XFPs has been introduced several years ago (Weber et al., 2011), as has been the usage of AAV vectors (Cai et al., 2013) or electroporation (Loulier et al., 2014) to achieve multicolor neuronal labeling, and the modeling of the relation between copy number and color combinations (Kobiler et al., 2010).

We are fully aware of all of these publications and, of course, have cited them in our initial submission. The major improvement introduced in this study is the enhanced expression with Tet-Off system. Furthermore, the result was qualitatively different. In the published methods, we could not clearly visualize fine axons without antibody staining. Therefore, they were not sufficiently powerful for large-scale analysis. However, with our new Tetbow strategy, we could trace millimeter-long axons after tissue clearing. We thus believe that our method is a major breakthrough toward large-scale neuronal tracing.

Second, beyond the images presented in the article, there is no demonstration of a usage of the Tetbow approach to trace connectivity. An application of these strategies to study some aspects of brain circuitry is essential to evaluate their usefulness. In particular it appears uncertain that labeled neuronal processes can be followed in their entirety in neural tissue samples.

To address this point, we have performed new experiments, which are now in Figure 8. At the very least, Tetbow-labelled axons were found in entire regions of the olfactory cortex (olfactory tubercle, piriform cortex, cortical amygdala, and lateral entorhinal cortex).

Third, due to incomplete description or inappropriate evaluation procedures, the actual improvement brought by Tetbow appears uncertain. For instance, it is unclear if the number of samples analyzed in Figure 1 is sufficient to minimize discrepancies among the different brains analyzed, and between different sections of a given brain.

The old Figure 1 only contains simulation results. If this reviewer meant Figure 2, we analyzed three brains each. In our new Figure 3F, we indicate three different brain samples separately in circles. There was no discrepancy among samples.

The authors also claim that their scheme is simpler that Brainbow, but in practice this is hardly the case: with Tetbow, a total of 4 plasmids or AAV vectors must be introduced in neurons of interest, while with Brainbow two plasmids (Brainbow et al., 2014) or 2 AAVs (AAV no. 1 and no. 2, Cai et al., 2013) are sufficient. In addition, with Tetbow, careful titration of the different color vectors needs to be performed for each fresh vector preparation, while expression of different XFPs is intrinsically balanced in Brainbow transgenes, making them inherently more reproducible.

We examined whether careful titration of the different color vectors are critical in our method. Both in simulation based on Poisson distribution (Figure 1G) and actual experiments (Figure 3 for plasmids; Figure 7 for AAVs); a 3-5-fold difference did not produce a big difference. These new sets of experiments demonstrate that Tetbow is highly reproducible, irrespective of small changes in plasmid amount or virus titers.

Of course, when the copy number is too high, the color variation will decrease, but this is also the case for Brainbow.

Concerning the strategy based on protein tags (Figure 5), one cannot assess how deep synthetic fluorochromes labels diffuse within tissue sections: does this scheme really enable to label more than just the first few 10th of micrometers of tissue samples?

Compared to antibodies, which are really huge, the molecular weights of chemical tag substrates are very small. To demonstrate efficient penetration or chemical tags, we stained 1mm-thick brain slices expressing chemical tags in layer 2/3 neurons. After staining with fluorescent substrates, the entire thickness was clearly visualized with fluorescent labels (Figure 6—figure supplement 2).

Finally, are Tetbow protein tags (Figure 5) and AAVs (Figure 6) more efficient in terms of expression level and color contrast than standard vectors? It seems that most cells in Figure 5 and Figure 6 coexpress all three XFPs and that the color contrast in these samples is low.

In terms of color variation and consistency, they are basically the same. In the chemical tag strategy, one protein tag reacts with just one substrate molecule. Thus, the brightness does not improve a lot, compared to the fluorescent proteins. More spectral variation is one advantage, as is already mentioned in the Discussion section. AAV can indeed enhance the expression level when compared to plasmids. This point is now mentioned in the Results section.

As the expression levels of XFPs are extremely high with the AAV Tetbow, XFPs start to aggregate and neurons start to die several weeks after virus injection. We sacrificed animals 2-4 weeks after injection, before these morphological damages emerge. This point is also clarified in the text.

[Editors' note: the author responses to the re-review follow.]

The manuscript has been improved but there are some remaining issues that need to be addressed before acceptance, as outlined below:While the reviewers were very positive about the extensive revision, they concluded after extended discussion that they would strongly recommend to address the following remaining issues. These will likely not require an additional consultation with the reviewers after submission of the final revision.

We thank all the reviewers for careful and constructive reviews.

- The text needs substantial revision along the suggestions of reviewers 2 and 3, especially to properly reference the literature and remove ambiguous statements.

According to the reviewers’ suggestions, we have amended the text.

- We recommend describing the quality of terminal axonal branch labeling, since this is considered a key benchmark of this set of methods.

We agree that it is a major interest for the readership in neuroscience community. However, it is almost impossible to know whether the labeled terminal are the real terminals or those that have been interrupted by technical reasons. This is true for any tracing techniques including EM-based connectomics. Therefore, there is no definitive tool to distinguish these two possibilities. Instead, we have clarified the following two points in the revised text.

1) We clarified the criteria as to how we determined the terminal of the tracing. In our analysis in Figure 8, we terminated tracing when axon labeling was terminated or interrupted by small unlabeled gaps. For the second situation, we cannot be 100% sure which ones are connected to each other. In this sense, our tracing was performed at conservative criteria and showing underestimates. This point is now clarified in Materilas and methods section and Figure legends.

2) Nevertheless, our tracing quality was at a similar level compared to Igarashi et al., 2012 and Ghosh et al., 2011 (Discussion section). We discussed the limitations and future challenged in a new subsection of Discussion section.

Reviewer #1:This manuscript is a revision of an earlier submission. The authors have taken serious steps to address the issues raised before and have in my view substantially improved the manuscript. The case for Tetbow being a significant step forward for multi-color neuronal labeling has been made more clearly.In particular, I think the added analyses of color assessment, comparison of fluorescence intensity to Brainbow; and importantly the analysis of color constancy over long distances (Figure 8 and associated supplements) are valuable and relevant.

We agree that the new sets of experiments have much strengthened the argument of this paper. We thank all the reviewers for constructive review process.

Reviewer #2:The revised manuscript, Tetbow: bright mutlicolor labeling of neuronal circuits with fluorescent proteins and chemical tags" by Sakaguchi et al., addresses some of the points raised by the reviewers in the previous submission however, one significant concern remains.This methods paper adds to the existing toolkit for multicolor labeling of neurons. The tTA-TRE based reagents described in this manuscript appear to have improved brightness over the traditional Brainbow methods. While such a color palette would certainly be useful for answering certain neuroanatomical questions the revised manuscript still fails to address the primary concern that was raised in the original submission – whether such a multicolor labeling approach would be useful for complete neuronal reconstructions. Will the use of multicolor labeling permit reconstruction of entire neurons at higher densities as the authors suggest, i.e. is it an improvement over existing methods that simply use sparse and bright neuronal labeling with a limited color-set.Figure 8 and Figure 8—figure supplement 2 in this revised submission address this question. The data presented clearly show that the trajectories of the main axon of multiple M/T neurons can be traced. This might be useful if the goal was to identify the primary brain areas targeted by these neurons. But it is not at all clear from the data shown whether complete reconstructions of neurons would be feasible and therefore prompts the question if Tetbow labeling is bright enough to trace axonal arbors in entirety. Reconstructions that aim to trace out only the main axon and the first order collaterals are already possible even in Thy1-GFP transgenic animals (see for instance Guo et al., (2017)). The authors would have definitely compared their traced neurons to the Mitral/Tufted cell reconstructions presented in Igarashi, 2012 or Ghosh, 2011 (both articles are cited in the manuscript). Is it possible to get similar level of completeness using the Tetbow approach?

In the conventional light microscopy, imaging resolution is limited, particularly along z-axis. Therefore, it is often difficult to distinguish two fibers crossing over along z-axis. Multicolor labeling is more reliable for tracing at longer distances, as has been quantitatively demonstrated in Figure 8—figure supplement 1.

As detailed above, it is almost impossible to know by any methods where the real terminal is. What we can say for now is that we traced axons with very conservative criteria, and our tracing performance was at a similar level compared to Igarashi et al., 2012 and Ghosh et al., 2011. We could find labeled axons in all the areas of the olfactory cortices we examined, including the olfactory tubercle, piriform cortex, cortical amygdala, and lateral entorhinal cortex (now in Discussion section). We could also find several axon collaterals in each mitral cell, which was comparable to these earlier studies. Of course, the single neuron labeling approach (Igarashi et al., 2012) has some advantages, e.g., robustness against interrupted labeling. On the other hand, our multicolor labeling approach has an advantage for the comparison of multiple neurons in the same animal. We could more clearly demonstrated heterogeneity of axonal projection patterns than earlier works. This point is now more extensively discussed in the revised text (Discussion section).

Reviewer #3:Sakaguchi et al., present a revised version of their article on the development of an enhanced multicolor neuronal labeling toolbox termed "Tetbow". They have significantly strengthened their study, […] The tools presented in the manuscript and the efforts made at characterizing optimal labeling conditions will be of interest for the neuroscience community.

We appreciate favorable comments on our revision.

My main comment about this new version of the article concerns the presentation of the results, which requires significant revision of the text for the following reasons:-Several concepts presented in the article are not novel per se and are simply improved and more deeply explored than in the studies that introduced them. The true interest of the article lies in these improvements and optimizations, which will undoubtedly be useful to the community, not in the rediscovery of previous ideas. This is for instance the case of multicolor labeling with mixtures of distinct single-color viral vectors (Chan et al., 2017; Weber et al., 2011) or plasmids (Siddiqi et al., 2014), modeling of the relationship between copy number and color (Kobiler et al., 2010), usage of the Tet-Off system to amplify expression in a multicolor context (Chan et al., 2017 and other studies)…When introducing these concepts, the text should state this explicitly and refer to the related articles. It is with respect to these recent studies that the new tools should be judged, not only relative to former Cre-dependent Brainbow approaches.

We totally agree with this reviewer. This is the first application of multicolor labeling in cleared tissues without further antibody labeling, but conceptually just the combination of existing ideas. We also apologize for our oversight of Siddiqi et al., 2014; this is now cited (subsection “A trade-off between expression levels and color variation”). We have also deleted the following sentence in the Results section: “We therefore tried to develop a multicolor labeling method for vector-mediated gene transfer”. Chan et al., 2017 successfully utilized Tet-Off system to sparsen the expression of XFPs, but did not show any quantitative evidence for amplified expression for XFPs. We therefore only cited Madisen, 2015 and Sadakane, 2015, which demonstrated improved fluorescence intensity quantitatively.

- The discussion should also include a section on the drawbacks of Tetbow and possible difficulties of this approach, such as the necessity to carefully titrate the different RGB vectors to equilibrate their concentration, batch-to-batch variations, and whether it is compatible with strategies to sparsen expression e.g. as in Chan et al., 2017.

We agree that we should state the limitations of our method in a section of Discussion section. As detailed in the previous rebuttal letter, batch-to-batch variation is not a major issue. Current limitations include filling performance and color consistency that would be required for automatic tracing in the future. These issues are now discussed with possible future solutions (subsection “Single axon tracing for long-range axonal projections”). The utility of the Tet-Off system for sparse expression is discussed in subsection “Bright multicolor labeling of neurons using Tet-Off system”.

-Some statements are repeated unnecessarily throughout the paper (e.g. the known fact that Cre recombination is not required for combinatorial labeling).

This statement has been entirely removed from the Discussion section, as suggested. Throughout the text, we tried to minimize redundant statements.

- Several sentences are quite imprecise, some conclusions are overstated and there are also many typos that need to be corrected.

We apologize for any difficulty in understanding due to poor descriptions and typos that appeared in the previous version. We have carefully proofread this revision.